# Coupled Oscillatory Recurrent Neural Network (coRNN): An accurate and (gradient) stable architecture for learning long time dependencies

**T. Konstantin Rusch**
Seminar for Applied Mathematics (SAM)
Department of Mathematics
ETH Zürich
Zürich, 8092, Switzerland
`trusch@ethz.ch`

**Siddhartha Mishra**
Seminar for Applied Mathematics (SAM)
Department of Mathematics
ETH Zürich
Zürich, 8092, Switzerland
`smishra@ethz.ch`

## Abstract

Circuits of biological neurons, such as in the functional parts of the brain can be modeled as networks of coupled oscillators. Inspired by the ability of these systems to express a rich set of outputs while keeping (gradients of) state variables bounded, we propose a novel architecture for recurrent neural networks. Our proposed RNN is based on a time-discretization of a system of second-order ordinary differential equations, modeling networks of controlled nonlinear oscillators. We prove precise bounds on the gradients of the hidden states, leading to the mitigation of the exploding and vanishing gradient problem for this RNN. Experiments show that the proposed RNN is comparable in performance to the state of the art on a variety of benchmarks, demonstrating the potential of this architecture to provide stable and accurate RNNs for processing complex sequential data.

## 1 Introduction

Recurrent neural networks (RNNs) have achieved tremendous success in a variety of tasks involving sequential (time series) inputs and outputs, ranging from speech recognition to computer vision and natural language processing, among others. However, it is well known that training RNNs to process inputs over long time scales (input sequences) is notoriously hard on account of the so-called *exploding and vanishing gradient problem (EVGP)* (Pascanu et al., 2013), which stems from the fact that the well-established BPTT algorithm for training RNNs requires computing products of gradients (Jacobians) of the underlying hidden states over very long time scales. Consequently, the overall gradient can grow (to infinity) or decay (to zero) exponentially fast with respect to the number of recurrent interactions.

A variety of approaches have been suggested to mitigate the exploding and vanishing gradient problem. These include adding *gating mechanisms* to the RNN in order to control the flow of information in the network, leading to architectures such as *long short-term memory* (LSTM) (Hochreiter & Schmidhuber, 1997) and *gated recurring units* (GRU) (Cho et al., 2014), that can overcome the vanishing gradient problem on account of the underlying additive structure. However, the gradients might still explode and learning very long term dependencies remains a challenge (Li et al., 2018). Another popular approach for handling the EVGP is to *constrain* the structure of underlying recurrent weight matrices by requiring them to be orthogonal (unitary), leading to the so-called *orthogonal RNNs* (Henaff et al., 2016; Arjovsky et al., 2016; Wisdom et al., 2016; Kerg et al., 2019) and references therein. By construction, the resulting Jacobians have eigen- and singular-spectra with unit norm, alleviating the EVGP. However as pointed out by Kerg et al. (2019), imposing such constraints on the recurrent matrices may lead to a significant loss of *expressivity* of the RNN resulting in inadequate performance on realistic tasks.

In this article, we adopt a different approach, based on observation that *coupled networks of controlled non-linear forced and damped oscillators*, that arise in many physical, engineering and biological

systems, such as networks of biological neurons, do seem to ensure expressive representations while constraining the dynamics of state variables and their gradients. This motivates us to propose a novel architecture for RNNs, based on time-discretizations of second-order systems of non-linear ordinary differential equations (ODEs) (1) that model coupled oscillators. Under verifiable hypotheses, we are able to *rigorously prove precise bounds on the hidden states of these RNNs and their gradients, enabling a possible solution of the exploding and vanishing gradient problem*, while demonstrating through benchmark numerical experiments, that the resulting system still retains sufficient expressivity, i.e. ability to process complex inputs, with a competitive performance, with respect to the state of the art, on a variety of sequential learning tasks.

## 2 THE PROPOSED RNN

Our proposed RNN is based on the following second-order system of ODEs,

$$\mathbf{y}'' = \sigma\left(\mathbf{W}\mathbf{y} + \mathcal{W}\mathbf{y}' + \mathbf{V}\mathbf{u} + \mathbf{b}\right) - \gamma\mathbf{y} - \epsilon\mathbf{y}'. \tag{1}$$

Here, $t \in [0, 1]$ is the (continuous) time variable, $\mathbf{u} = \mathbf{u}(t) \in \mathbb{R}^d$ is the time-dependent *input signal*, $\mathbf{y} = \mathbf{y}(t) \in \mathbb{R}^m$ is the *hidden state* of the RNN with $\mathbf{W}, \mathcal{W} \in \mathbb{R}^{m \times m}$, $\mathbf{V} \in \mathbb{R}^{m \times d}$ are weight matrices, $\mathbf{b} \in \mathbb{R}^m$ is the bias vector and $0 < \gamma, \epsilon$ are parameters, representing oscillation frequency and the amount of damping (friction) in the system, respectively. $\sigma : \mathbb{R} \mapsto \mathbb{R}$ is the *activation function*, set to $\sigma(u) = \tanh(u)$ here. By introducing the so-called *velocity* variable $\mathbf{z} = \mathbf{y}'(t) \in \mathbb{R}^m$, we rewrite (1) as the first-order system:

$$\mathbf{y}' = \mathbf{z}, \quad \mathbf{z}' = \sigma\left(\mathbf{W}\mathbf{y} + \mathcal{W}\mathbf{z} + \mathbf{V}\mathbf{u} + \mathbf{b}\right) - \gamma\mathbf{y} - \epsilon\mathbf{z}. \tag{2}$$

We fix a timestep $0 < \Delta t < 1$ and define our proposed RNN hidden states at time $t_n = n\Delta t \in [0, 1]$ (while omitting the affine output state) as the following IMEX (implicit-explicit) discretization of the first order system (2):

$$\begin{aligned}
\mathbf{y}_n &= \mathbf{y}_{n-1} + \Delta t \mathbf{z}_n, \\
\mathbf{z}_n &= \mathbf{z}_{n-1} + \Delta t \sigma\left(\mathbf{W}\mathbf{y}_{n-1} + \mathcal{W}\mathbf{z}_{n-1} + \mathbf{V}\mathbf{u}_n + \mathbf{b}\right) - \Delta t\gamma\mathbf{y}_{n-1} - \Delta t\epsilon\mathbf{z}_{\bar{n}},
\end{aligned} \tag{3}$$

with either $\bar{n} = n$ or $\bar{n} = n - 1$. Note that the only difference in the two versions of the RNN (3) lies in the implicit ($\bar{n} = n$) or explicit ($\bar{n} = n - 1$) treatment of the damping term $-\epsilon\mathbf{z}$ in (2), whereas both versions retain the implicit treatment of the first equation in (2).

**Motivation and background.** To see that the underlying ODE (2) models a *coupled network of controlled forced and damped nonlinear oscillators*, we start with the single neuron (scalar) case by setting $d = m = 1$ in (1) and assume an identity activation function $\sigma(x) = x$. Setting $\mathbf{W} = \mathcal{W} = \mathbf{V} = \mathbf{b} = \epsilon = 0$ leads to the simple ODE, $\mathbf{y}'' + \gamma\mathbf{y} = 0$, which exactly models *simple harmonic motion* with *frequency* $\gamma$, for instance that of a mass attached to a spring (Guckenheimer & Holmes, 1990). Letting $\epsilon > 0$ in (1) adds *damping* or friction to the system (Guckenheimer & Holmes, 1990). Then, by introducing non-zero $\mathbf{V}$ in (1), we drive the system with a driving force proportional to the input signal $\mathbf{u}(t)$. The parameters $\mathbf{V}, \mathbf{b}$ modulate the effect of the driving force, $\mathbf{W}$ controls the frequency of oscillations and $\mathcal{W}$ the amount of damping in the system. Finally, the $\tanh$ activation mediates a non-linear response in the oscillator. In the coupled network (2) with $m > 1$, each neuron updates its hidden state based on the input signal as well as information from other neurons. The diagonal entries of $\mathbf{W}$ (and the scalar hyperparameter $\gamma$) control the frequency whereas the diagonal entries of $\mathcal{W}$ (and the hyperparameter $\epsilon$) determine the amount of damping for each neuron, respectively, whereas the non-diagonal entries of these matrices modulate interactions between neurons. Hence, given this behavior of the underlying ODE (2), we term the RNN (3) as a *coupled oscillatory Recurrent Neural Network* (coRNN).

The dynamics of the ODE (2) (and the RNN (3)) for a single neuron are relatively straightforward. As we illustrate in Fig. 6 of supplementary material **SM**§C, input signals drive the generation of (superpositions of) oscillatory wave-forms, whose amplitude and (multiple) frequencies are controlled by the tunable parameters $\mathbf{W}, \mathcal{W}, \mathbf{V}, \mathbf{b}$. Adding a $\tanh$ activation does not change these dynamics much. This is in contrast to truncating $\tanh$ to leading non-linear order by setting $\sigma(x) = x - x^3/3$, which yields a Duffing type oscillator that is characterized by chaotic behavior (Guckenheimer & Holmes, 1990). Adding interactions between neurons leads to further accentuation of this generation of superposed wave forms (see Fig. 6 in **SM**§C) and even with very simple network topologies, one

sees the emergence of non-trivial non-oscillatory hidden states from oscillatory inputs. In practice, a network of a large number of neurons is used and can lead to extremely rich global dynamics. Hence, we argue that the ability of a network of (forced, driven) oscillators to access a very rich set of output states may lead to high expressivity of the system, allowing it to approximate outputs from complicated sequential inputs.

Oscillator networks are ubiquitous in nature and in engineering systems (Guckenheimer & Holmes, 1990; Strogatz, 2015) with canonical examples being pendulums (classical mechanics), business cycles (economics), heartbeat (biology) for single oscillators and electrical circuits for networks of oscillators. Our motivating examples arise in neurobiology, where individual biological neurons can be viewed as oscillators with periodic spiking and firing of the action potential. Moreover, functional circuits of the brain, such as cortical columns and prefrontal-striatal-hippocampal circuits, are being increasingly interpreted by networks of oscillatory neurons, see Stiefel & Ermentrout (2016) for an overview. Following well-established paths in machine learning, such as for convolutional neural networks (LeCun et al., 2015), our focus here is to abstract the essence of functional brain circuits being networks of oscillators and design an RNN based on much simpler mechanistic systems, such as those modeled by (2), while ignoring the complicated biological details of neural function.

**Related work.** There is an increasing trend of basing RNN architectures on ODEs and dynamical systems. These approaches can roughly be classified into two branches, namely RNNs based on discretized ODEs and continuous-time RNNs. Examples of continuous-time approaches include neural ODEs (Chen et al., 2018) with ODE-RNNs (Rubanova et al., 2019) as its recurrent extension as well as E (2017) and references therein, to name just a few. We focus, however, in this article on an ODE-inspired discrete-time RNN, as the proposed coRNN is derived from a discretization of the ODE (1). A good example for a discrete-time ODE-based RNNs is the so-called *anti-symmetric* RNN of Chang et al. (2019), where the RNN architecture is based on a stable ODE resulting from a skew-symmetric hidden weight matrix, thus constraining the stable (gradient) dynamics of the network. This approach has much in common with previously mentioned unitary/orthogonal/non-normal RNNs in constraining the structure of the hidden-to-hidden layer weight matrices. However, adding such strong constraints might reduce expressivity of the resulting RNN and might lead to inadequate performance on complex tasks. In contrast to these approaches, our proposed coRNN does not explicitly constrain the weight matrices but relies on the dynamics of the underlying ODE (and the IMEX discretization (3)), to provide gradient stability. Moreover, no gating mechanisms as in LSTMs/GRUs are used in the current version of coRNN. There is also an increasing interest in designing *hybrid* methods, which use a discretization of an ODE (in particular a Hamiltonian system) in order to learn the continuous representation of the data, see for instance Greydanus et al. (2019); Chen et al. (2020). Overall, our approach here differs from these papers in our use of networks of oscillators to build the RNN.

## 3 RIGOROUS ANALYSIS OF THE PROPOSED RNN

An attractive feature of the underlying ODE system (2) lies in the fact that the resulting hidden states (and their gradients) are bounded (see **SM**§D for precise statements and proofs). Hence, one can expect that a suitable discretization of the ODE (2) that preserves these bounds will not have exploding gradients. We claim that one such *structure preserving discretization* is given by the IMEX discretization that results in the RNN (3) and proceed to derive bounds on this RNN below.

Following standard practice we set $\mathbf{y}(0) = \mathbf{z}(0) = 0$ and purely for the simplicity of exposition, we set the control parameters, $\epsilon = \gamma = 1$ and $\bar{n} = n$ in (3) leading to,

$$
\begin{aligned}
\mathbf{y}_n &= \mathbf{y}_{n-1} + \Delta t \mathbf{z}_n, \\
\mathbf{z}_n &= \frac{\mathbf{z}_{n-1}}{1+\Delta t} + \frac{\Delta t}{1+\Delta t}\sigma(\mathbf{A}_{n-1}) - \frac{\Delta t}{1+\Delta t}\mathbf{y}_{n-1}, \quad \mathbf{A}_{n-1} := \mathbf{W}\mathbf{y}_{n-1} + \mathcal{W}\mathbf{z}_{n-1} + \mathbf{V}\mathbf{u}_n + \mathbf{b}.
\end{aligned}
\tag{4}
$$

Analogous results and proofs for the case where $\bar{n} = n - 1$ and for general values of $\epsilon, \gamma$ are provided in **SM**§F.

**Bounds on the hidden states.** As with the underlying ODE (2), the hidden states of the RNN (3) are bounded, i.e.

**Proposition 3.1** *Let* $\mathbf{y}_n, \mathbf{z}_n$ *be the hidden states of the RNN* (4) *for* $1 \leq n \leq N$, *then the hidden states satisfy the following (energy) bounds:*

$$\mathbf{y}_n^\top \mathbf{y}_n + \mathbf{z}_n^\top \mathbf{z}_n \leq nm\Delta t = mt_n \leq m. \tag{5}$$

The proof of the *energy* bound (5) is provided in **SM**§E.1 and a straightforward variant of the proof (see **SM**§E.2) yields an estimate on the sensitivity of the hidden states to changing inputs. As with the underlying ODE (see **SM**§D) , this bound *rules out chaotic behavior of hidden states*.

**Bounds on hidden state gradients.**    We train the RNN (3) to minimize the loss function,

$$\mathcal{E} := \frac{1}{N} \sum_{n=1}^N \mathcal{E}_n, \quad \mathcal{E}_n = \frac{1}{2} \|\mathbf{y}_n - \bar{\mathbf{y}}_n\|_2^2, \tag{6}$$

with $\bar{\mathbf{y}}$ being the underlying ground truth (training data). During training, we compute gradients of the loss function (6) with respect to the weights and biases $\boldsymbol{\Theta} = [\mathbf{W}, \mathcal{W}, \mathbf{V}, \mathbf{b}]$, i.e.

$$\frac{\partial \mathcal{E}}{\partial \theta} = \frac{1}{N} \sum_{n=1}^N \frac{\partial \mathcal{E}_n}{\partial \theta}, \quad \forall \, \theta \in \boldsymbol{\Theta}. \tag{7}$$

**Proposition 3.2** *Let* $\mathbf{y}_n, \mathbf{z}_n$ *be the hidden states generated by the RNN* (4). *We assume that the time step* $\Delta t << 1$ *can be chosen such that,*

$$\max \left\{ \frac{\Delta t(1 + \|\mathbf{W}\|_\infty)}{1 + \Delta t}, \frac{\Delta t \|\mathcal{W}\|_\infty}{1 + \Delta t} \right\} = \eta \leq \Delta t^r, \quad \frac{1}{2} \leq r \leq 1. \tag{8}$$

*Denoting* $\delta = \frac{1}{1+\Delta t}$, *the gradient of the loss function* $\mathcal{E}$ (6) *with respect to any parameter* $\theta \in \boldsymbol{\Theta}$ *is bounded as,*

$$\left| \frac{\partial \mathcal{E}}{\partial \theta} \right| \leq \frac{3}{2} \left( m + \bar{Y}\sqrt{m} \right), \tag{9}$$

*with* $\bar{Y} = \max_{1 \leq n \leq N} \|\bar{\mathbf{y}}_n\|_\infty$ *be a bound on the underlying training data.*

*Sketch of the proof.* Denoting $\mathbf{X}_n = [\mathbf{y}_n, \mathbf{z}_n]$, we can apply the chain rule repeatedly (for instance as in Pascanu et al. (2013)) to obtain,

$$\frac{\partial \mathcal{E}_n}{\partial \theta} = \sum_{1 \leq k \leq n} \underbrace{\frac{\partial \mathcal{E}_n}{\partial \mathbf{X}_n} \frac{\partial \mathbf{X}_n}{\partial \mathbf{X}_k} \frac{\partial^+ \mathbf{X}_k}{\partial \theta}}_{\frac{\partial \mathcal{E}_n^{(k)}}{\partial \theta}}. \tag{10}$$

Here, the notation $\frac{\partial^+ \mathbf{X}_k}{\partial \theta}$ refers to taking the partial derivative of $\mathbf{X}_k$ with respect to the parameter $\theta$, while keeping the other arguments constant. This quantity can be readily calculated from the structure of the RNN (4) and is presented in the detailed proof provided in **SM**§E.3. From (6), we can directly compute that $\frac{\partial \mathcal{E}_n}{\partial \mathbf{X}_n} = [\mathbf{y}_n - \bar{\mathbf{y}}_n, 0]$.

Repeated application of the chain rule and a direct calculation with (4) yields,

$$\frac{\partial \mathbf{X}_n}{\partial \mathbf{X}_k} = \prod_{k < i \leq n} \frac{\partial \mathbf{X}_i}{\partial \mathbf{X}_{i-1}}, \quad \frac{\partial \mathbf{X}_i}{\partial \mathbf{X}_{i-1}} = \begin{bmatrix} \mathbf{I} + \Delta t \mathbf{B}_{i-1} & \Delta t \mathbf{C}_{i-1} \\ \mathbf{B}_{i-1} & \mathbf{C}_{i-1} \end{bmatrix}, \tag{11}$$

where I is the identity matrix and

$$\mathbf{B}_{i-1} = \delta \Delta t \left( \mathrm{diag}(\sigma'(\mathbf{A}_{i-1})) \mathbf{W} - \mathbf{I} \right), \quad \mathbf{C}_{i-1} = \delta \left( \mathbf{I} + \Delta t \, \mathrm{diag}(\sigma'(\mathbf{A}_{i-1})) \mathcal{W} \right). \tag{12}$$

It is straightforward to calculate using the assumption (8) that $\|\mathbf{B}_{i-1}\|_\infty < \eta$ and $\|\mathbf{C}_{i-1}\|_\infty \leq \eta + \delta$. Using the definitions of matrix norms and (8), we obtain:

$$\left\| \frac{\partial \mathbf{X}_i}{\partial \mathbf{X}_{i-1}} \right\|_\infty \leq \max \left( 1 + \Delta t(\|\mathbf{B}_{i-1}\|_\infty + \|\mathbf{C}_{i-1}\|_\infty), \|\mathbf{B}_{i-1}\|_\infty + \|\mathbf{C}_{i-1}\|_\infty \right)$$
$$\leq \max \left( 1 + \Delta t(\delta + 2\eta), \delta + 2\eta \right) \leq 1 + 3\Delta t^r. \tag{13}$$

Therefore, using (11), we have

$$\left\| \frac{\partial \mathbf{X}_n}{\partial \mathbf{X}_k} \right\|_\infty \leq \prod_{k < i \leq n} \left\| \frac{\partial \mathbf{X}_i}{\partial \mathbf{X}_{i-1}} \right\|_\infty \leq (1 + 3\Delta t^r)^{n-k} \approx 1 + 3(n - k)\Delta t^r. \tag{14}$$

Note that we have used an expansion around 1 and neglected terms of $\mathcal{O}(\Delta t^{2r})$ as $\Delta t << 1$. We remark that the bound (13) is the *crux of our argument* about gradient control as we see from the structure of the RNN that the recurrent matrices have close to unit norm. The detailed proof is presented in **SM**§E.3. As the entire gradient of the loss function (6), with respect to the weights and biases of the network, is bounded above in (9), the *exploding gradient problem* is mitigated for this RNN.

**On the vanishing gradient problem.** The vanishing gradient problem (Pascanu et al., 2013) arises if $\left| \frac{\partial \mathcal{E}_n^{(k)}}{\partial \theta} \right|$, defined in (10), $\to 0$ exponentially fast in $k$, for $k << n$ (long-term dependencies). In that case, the RNN does not have long-term memory, as the contribution of the $k$-th hidden state to error at time step $t_n$ is infinitesimally small. We already see from (14) that $\left\| \frac{\partial \mathbf{X}_n}{\partial \mathbf{X}_k} \right\|_\infty \approx 1$ (independently of k). Thus, we should not expect the products in (10) to decay fast. In fact, we will provide a much more precise characterization of this gradient. To this end, we introduce the following *order*-notation,

$$\beta = \mathcal{O}(\alpha), \text{for } \alpha, \beta \in \mathbb{R}_+ \quad \text{if there exists constants } \overline{C}, \underline{C} \text{ such that } \underline{C}\alpha \leq \beta \leq \overline{C}\alpha.$$
$$\mathbf{M} = \mathcal{O}(\alpha), \text{for } \mathbf{M} \in \mathbb{R}^{d_1 \times d_2}, \alpha \in \mathbb{R}_+ \quad \text{if there exists constant } \overline{C} \text{ such that } \|\mathbf{M}\| \leq \overline{C}\alpha. \tag{15}$$

For simplicity of notation, we will also set $\bar{\mathbf{y}}_n = \mathbf{u}_n \equiv 0$, for all $n$, $\mathbf{b} = 0$ and $r = 1$ in (8) and we will only consider $\theta = \mathbf{W}_{i,j}$ for some $1 \leq i, j \leq m$ in the following proposition.

**Proposition 3.3** *Let $\mathbf{y}_n$ be the hidden states generated by the RNN (4). Under the assumption that $\mathbf{y}_n^i = \mathcal{O}(\sqrt{t_n})$, for all $1 \leq i \leq m$ and (8), the gradient for long-term dependencies satisfies,*

$$\frac{\partial \mathcal{E}_n^{(k)}}{\partial \theta} = \mathcal{O}\left(\hat{c}\delta\Delta t^{\frac{3}{2}}\right) + \mathcal{O}\left(\hat{c}\delta(1 + \delta)\Delta t^{\frac{5}{2}}\right) + \mathcal{O}(\Delta t^3), \ \hat{c} = sech^2\left(\sqrt{k\Delta t}(1 + \Delta t)\right), \ k << n. \tag{16}$$

This precise bound (16) on the gradient shows that although the gradient can be small, i.e $\mathcal{O}(\Delta t^{\frac{3}{2}})$, it is in fact *independent of $k$*, ensuring that long-term dependencies contribute to gradients at much later steps and mitigating the vanishing gradient problem. The detailed proof is presented in **SM**§E.5.

Summarizing, we see that the RNN (3) indeed satisfied similar bounds to the underlying ODE (2) that resulted in upper bounds on the hidden states and its gradients. However, the lower bound on the gradient (16) is due to the specific choice of this discretization and does not appear to have a continuous analogue, making the specific choice of discretization of (2) crucial for mitigating the vanishing gradient problem.

## 4    EXPERIMENTS

We present results on a variety of learning tasks with coRNN (3) with $\bar{n} = n - 1$, as this version resulted in marginally better performance than the version with $\bar{n} = n$. Details of the training procedure for each experiment can be found in **SM**§B. We wish to clarify here that we use a straightforward hyperparameter tuning protocol based on a validation set and do not use additional performance enhancing tools, such as dropout (Srivastava et al., 2014), gradient clipping (Pascanu et al., 2013) or batch normalization (Ioffe & Szegedy, 2015), which might further improve the performance of coRNNs.

**Adding problem.** We start with the well-known adding problem (Hochreiter & Schmidhuber, 1997), proposed to test the ability of an RNN to learn (very) long-term dependencies. The input is a two-dimensional sequence of length $T$, with the first dimension consisting of random numbers drawn from $\mathcal{U}([0, 1])$ and with two non-zero entries (both set to 1) in the second dimension, chosen at random locations, but one each in both halves of the sequence. The output is the sum of two numbers

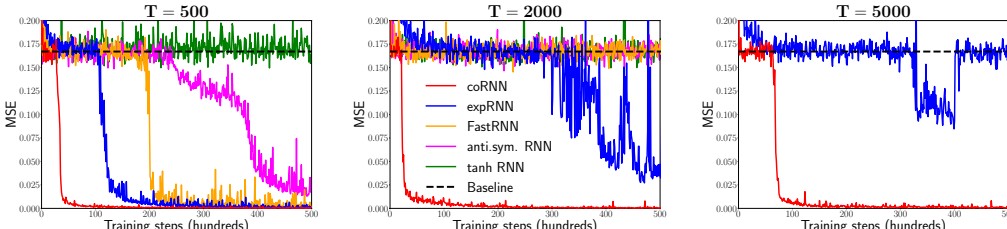

Figure 1: Results of the adding problem for coRNN, expRNN, FastRNN, anti.sym. RNN and tanh RNN based on three different sequence lengths $T$, i.e. $T = 500$, $T = 2000$ and $T = 5000$.

of the first dimension at positions, corresponding to the two 1 entries in the second dimension. We compare the proposed coRNN to three recently proposed RNNs, which were explicitly designed to learn LTDs, namely the FastRNN (Kusupati et al., 2018), the antisymmetric (anti.sym.) RNN (Chang et al., 2019) and the expRNN (Lezcano-Casado & Martínez-Rubio, 2019), and to a plain vanilla tanh RNN, with the goal of beating the baseline mean square error (MSE) of $0.167$ (which stems from the variance of the baseline output 1). All methods have 128 hidden units (dimensionality of the hidden state $\mathbf{y}$) and the same training protocol is used in all cases. Fig. 1 shows the results for different lengths $T$ of the input sequences. We can see that while the tanh RNN is not able to beat the baseline for any sequence length, the other methods successfully learn the adding task for $T = 500$. However, in this case, coRNN converges significantly faster and reaches a lower test MSE than other tested methods. When setting the length to the much more challenging case of $T = 2000$, we see that only coRNN and the expRNN beat the baseline. However, the expRNN fails to reach a desired test MSE of $0.01$ within training time. In order to further demonstrate the superiority of coRNN over recently proposed RNN architectures for learning LTDs, we consider the adding problem for $T = 5000$ and observe that coRNN converges very quickly even in this case, while expRNN fails to consistently beat the baseline. We thus conclude that the coRNN mitigates the vanishing/exploding gradient problem even for very long sequences.

Table 1: Test accuracies on sMNIST and psMNIST (we provide our own psMNIST result for the FastGRNN, as no official result for this task has been published so far).

| Model | sMNIST | psMNIST | # units | # params |
|---|---|---|---|---|
| uRNN (Arjovsky et al., 2016) | 95.1% | 91.4% | 512 | 9k |
| LSTM (Helfrich et al., 2018) | 98.9% | 92.9% | 256 | 270k |
| GRU (Chang et al., 2017) | 99.1% | 94.1% | 256 | 200k |
| anti.sym. RNN (Chang et al., 2019) | 98.0% | 95.8% | 128 | 10k |
| DTRIV$\infty$ (Casado, 2019) | 99.0% | 96.8% | 512 | 137k |
| FastGRNN (Kusupati et al., 2018) | 98.7% | 94.8% | 128 | 18k |
| **coRNN** (128 units) | 99.3% | 96.6% | 128 | 34k |
| **coRNN** (256 units) | **99.4%** | **97.3%** | 256 | 134k |

**Sequential (permuted) MNIST.** Sequential MNIST (sMNIST) (Le et al., 2015) is a benchmark for RNNs, in which the model is required to classify an MNIST (LeCun et al., 1998) digit one pixel at a time leading to a classification task with a sequence length of $T = 784$. In permuted sequential MNIST (psMNIST), a fixed random permutation is applied in order to increase the time-delay between interdependent pixels and to make the problem harder. In Table 1, we compare the test accuracy for coRNN on sMNIST and psMNIST with recently published best case results for other recurrent models, which were explicitly designed to solve long-term dependencies together with baselines corresponding to gated and unitary RNNs. To the best of our knowledge the proposed coRNN outperforms all single-layer recurrent architectures, published in the literature, for both the sMNIST and psMNIST. Moreover in Fig. 2, we present the performance (with respect to number of epochs) of different RNN architectures for psMNIST with the same fixed random permutation and the

same number of hidden units, i.e. 128. As seen from this figure, coRNN clearly outperforms the other architectures, some of which were explicitly designed to learn LTDs, handily for this permutation.

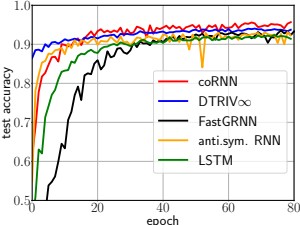 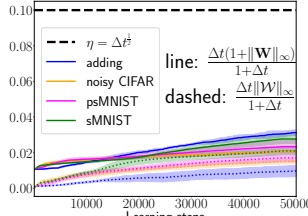 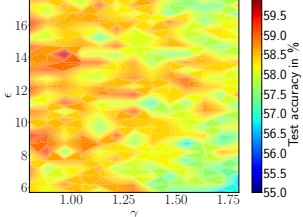

Figure 2: Performance on psM-NIST for different models, all with 128 hidden units and the same fixed random permutation.

Figure 3: Weight assumptions (8), with $r = \frac{1}{2}$, evaluated during training for all LTD experiments (mean and standard deviation of 10 different runs for each task).

Figure 4: Ablation study on the hyperparameters $\epsilon, \gamma$ in (3) using the noise padded CIFAR-10 experiment.

**Noise padded CIFAR-10.**   Another challenging test problem for learning LTDs is the recently proposed noise padded CIFAR-10 experiment by Chang et al. (2019), in which CIFAR-10 data points (Krizhevsky et al., 2009) are fed to the RNN row-wise and flattened along the channels resulting in sequences of length 32. To test the long term memory, entries of uniform random numbers are added such that the resulting sequences have a length of 1000, i.e. the last 968 entries of each sequence are only noise to distract the network. Table 2 shows the result for coRNN together with other recently published best case results. We observe that coRNN readily outperforms other RNN architectures on this benchmark, while requiring only 128 hidden units.

Table 2: Test accuracies on noise padded CIFAR-10.

| Model | test accuracy | # units | # params |
|---|---|---|---|
| LSTM (Kag et al., 2020) | 11.6% | 128 | 64k |
| Incremental RNN (Kag et al., 2020) | 54.5% | 128 | 12k |
| FastRNN (Kag et al., 2020) | 45.8% | 128 | 16k |
| anti.sym. RNN (Chang et al., 2019) | 48.3% | 256 | 36k |
| Gated anti.sym. RNN (Chang et al., 2019) | 54.7% | 256 | 37k |
| Lipschitz RNN (Erichson et al., 2020) | 55.2% | 256 | 134k |
| **coRNN** | **59.0**% | 128 | 46k |

**Human activity recognition.**   This experiment is based on the human activity recognition data set provided by Anguita et al. (2012). The data set is a collection of tracked human activities, which were measured by an accelerometer and gyroscope on a Samsung Galaxy S3 smartphone. Six activities were binarized to obtain two merged classes {Sitting, Laying, Walking_Upstairs} and {Standing, Walking, Walking_Downstairs}, leading to the HAR-2 data set, which was first proposed in Kusupati et al. (2018). Table 3 shows the result for coRNN together with other very recently published best case results on the same data set. We can see that coRNN readily outperforms all other methods. We also ran this experiment on a *tiny coRNN* with very few parameters, i.e. only 1k. We can see that even in this case, the tiny coRNN beats all baselines. We thus conclude that coRNN can efficiently be used on resource-constrained IoT micro-controllers.

**IMDB sentiment analysis.**   The IMDB data set (Maas et al., 2011) is a collection of 50k movie reviews, where 25k reviews are used for training (with 7.5k of these reviews used for validating) and 25k reviews are used for testing. The aim of this binary sentiment classification task is to decide whether a movie review is positive or negative. We follow the standard procedure by initializing the word embedding with pretrained 100d GloVe (Pennington et al., 2014) vectors and restrict the

Table 3: Test accuracies on HAR-2.

| Model | test accuracy | # units | # params |
|-------|---------------|---------|----------|
| GRU (Kusupati et al., 2018) | 93.6% | 75 | 19k |
| LSTM (Kag et al., 2020) | 93.7% | 64 | 16k |
| FastRNN (Kusupati et al., 2018) | 94.5% | 80 | 7k |
| FastGRNN (Kusupati et al., 2018) | 95.6% | 80 | 7k |
| anti.sym. RNN (Kag et al., 2020) | 93.2% | 120 | 8k |
| incremental RNN (Kag et al., 2020) | 96.3% | 64 | 4k |
| **coRNN** | **97.2**% | 64 | 9k |
| *tiny coRNN* | 96.5% | 20 | 1k |

dictionary to 25k words. Table 4 shows the results for coRNN and other recently published models, which are trained similarly and have the same number of hidden units, i.e. 128. We can see that coRNN compares favorable with gated baselines (which are known to perform very well on this task), while at the same time requiring significantly less parameters.

Table 4: Test accuracies on IMDB.

| Model | test accuracy | # units | # params |
|-------|---------------|---------|----------|
| LSTM (Campos et al., 2018) | 86.8% | 128 | 220k |
| Skip LSTM(Campos et al., 2018) | 86.6% | 128 | 220k |
| GRU (Campos et al., 2018) | 86.2% | 128 | 164k |
| Skip GRU (Campos et al., 2018) | 86.6% | 128 | 164k |
| ReLU GRU (Dey & Salemt, 2017) | 84.8% | 128 | 99k |
| **coRNN** | **87.4**% | 128 | 46k |

**Further experimental results.** To shed further light on the performance of coRNN, we consider the following issues. First, the theory suggested that coRNN mitigates the exploding/vanishing gradient problem as long as the assumptions (8) on the time step $\Delta t$ and weight matrices $\mathbf{W}, \mathcal{W}$ hold. Clearly one can choose a suitable $\Delta t$ to enforce (8) before training, but do these assumptions remain valid during training? In **SM**§E.4, we argue, based on worst-case estimates, that the assumptions will remain valid for possibly a large number of training steps. More pertinently, we can verify experimentally that (8) holds during training. This is demonstrated in Fig. 3, where we show that (8) holds for all LTD tasks during training. Thus, the presented theory applies and one can expect control over hidden state gradients with coRNN. Next, we recall that the frequency parameter $\gamma$ and damping parameter $\epsilon$ play a role for coRNNs (see **SM**§F for the theoretical dependence and Table 8 for best performing values of $\epsilon, \gamma$ for each numerical experiment within the range considered in Table 7). How sensitive is the performance of coRNN to the choice of these 2 parameters? To investigate this dependence, we focus on the noise padded CIFAR-10 experiment and show the results of an *ablation study* in Fig. 4, where the test accuracy for different coRNNs based on a two dimensional hyperparameter grid $(\epsilon, \gamma) \in [0.8, 1.8] \times [5.7, 17, 7]$ (i.e., sufficiently large intervals around the best performing values of $\epsilon, \gamma$ from Table 8) is plotted. We observe from the figure that although there are reductions in test accuracy for non-optimal values of $(\epsilon, \gamma)$, there is no large variation and the performance is rather robust with respect to these hyperparameters. Finally, note that we follow standard practice and present best reported results with coRNN as well as other competing RNNs in order to compare the relative performance. However, it is natural to investigate the dependence of these *best* results on the random initial (before training) values of the weight matrices. To this end, in Table 5 of **SM**, we report the mean and standard deviation (over 10 retrainings) of the test accuracy with coRNN on various learning tasks and find that the mean value is comparable to the best reported value, with low standard deviations. This indicates further robustness of the performance of coRNNs.

Table 5: Distributional information (mean and standard deviation) on the results for each classification experiment presented in the paper based on 10 re-trainings of the best performing coRNN using random initialization of the trainable parameters.

| Experiment | Mean | Standard deviation |
|---|---|---|
| sMNIST (256 units) | 99.17% | 0.07% |
| psMNIST (256 units) | 96.10% | 1.20% |
| Noise padded CIFAR-10 | 58.56% | 0.35% |
| HAR-2 (64 units) | 96.01% | 0.53% |
| IMDB | 86.65% | 0.31% |

## 5 DISCUSSION

Inspired by many models in physics, biology and engineering, we proposed a novel RNN architecture (3) based on a model (1) of a *network of controlled forced and damped oscillators*. For this RNN, we rigorously showed that under verifiable hypotheses on the time step and weight matrices, the hidden states are bounded (5) and obtained precise bounds on the gradients (Jacobians) of the hidden states, (9) and (16). Thus by design, this architecture can mitigate the exploding and vanishing gradient problem (EVGP) for RNNs. We present a series of numerical experiments that include sequential image classification, activity recognition and sentiment analysis, to demonstrate that the proposed coRNN keeps hidden states and their gradients under control, while retaining sufficient expressivity to perform complex tasks. Thus, we provide a novel and promising strategy for designing RNN architectures that are motivated by the functioning of natural systems, have rigorous bounds on hidden state gradients and are robust, accurate, straightforward to train and cheap to evaluate.

This work can be extended in different directions. For instance in this article, we have mainly focused on the learning of tasks with long-term dependencies and observed that coRNNs are comparable in performance to the best published results in the literature. Given that coRNNs are built with networks of oscillators, it is natural to expect that they will perform very well on tasks with oscillatory inputs/outputs, such as the time series analysis of high-resolution biomedical data, for instance EEG (electroencephalography) and EMG (electromyography) data and seismic activity data from geoscience. This will be pursued in a follow-up article. Similarly, applications of coRNN to language modeling will be covered in future work.

However, it is essential to point out that coRNNs might not be suitable for every learning task involving sequential inputs/outputs. As a concrete example, we consider the problem of predicting time series corresponding to a chaotic dynamical system. We recall that by construction, the underlying ODE (2) (and the discretization (3)) do not allow for super-linear (in time) separation of trajectories for nearby inputs. Thus, we cannot expect that coRNNs will be effective at predicting chaotic time series and it is indeed investigated and demonstrated for a Lorenz-96 ODE in **SM**§A, where we observe that the coRNN is outperformed by LSTMs in the chaotic regime.

Our main theoretical focus in this paper was to demonstrate the possible mitigation of the exploding and vanishing gradient problem. On the other hand, we only provided some heuristics and numerical evidence on why the proposed RNN still has sufficient expressivity. A priori, it is natural to think that the proposed RNN architecture might introduce a strong bias towards oscillatory functions. However, as we argue in **SM**§C, the proposed coRNN can be significantly more expressive, as the damping, forcing and coupling of several oscillators modulates nonlinear response to yield a very rich and diverse set of output states. This is also evidenced by the ability of coRNNs to deal with many tasks in our numerical experiments, which do not have an explicit oscillatory structure. This sets the stage for a rigorous investigation of *universality* of the proposed coRNN architecture, as in the case of echo state networks in Grigoryeva & Ortega (2018). A possible approach would be to leverage the ability of the proposed RNN to convert general inputs into a rich set of superpositions of harmonics (oscillatory wave forms). Moreover, the proposed RNN was based on the simplest model of coupled oscillators (1). Much more detailed models of oscillators are available, particularly those that arise in the modeling of biological neurons, Stiefel & Ermentrout (2016) and references therein. An interesting variant of our proposed RNN would be to base the RNN architecture on these more elaborate models, resulting in analogues of the spiking neurons model of Maass (2001) for RNNs.

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

**Supplementary Material for:**
Coupled Oscillatory Recurrent Neural Network (coRNN): An accurate and (gradient) stable
architecture for learning long time dependencies



## A CHAOTIC TIME-SERIES PREDICTION.

According to proposition E.1, coRNN does not exhibit chaotic behavior by design. While this property is highly desirable for learning long-term dependencies (a slight perturbation of the input should not result in an unbounded perturbation of the prediction), it impairs the performance on tasks, where the network has to learn actual chaotic dynamics. To test this numerically, we consider the following version of the Lorenz 96 system: (Lorenz, 1996):

$$x'_j = (x_{i+1} - x_{i-2})x_{i-1} - x_i + F, \tag{17}$$

where $x_j \in \mathbb{R}$ for all $j = 1, \ldots, 5$ and $F$ is an external force controlling the level of chaos in the system. Fig. 5 shows a trajectory of the system (17) plotted on the $x_1 x_2$-plane for a small external

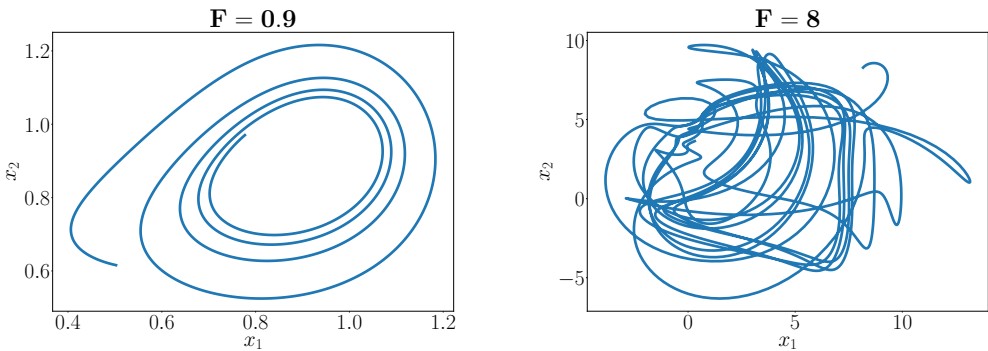

Figure 5: Exemplary $(x_1, x_2)$-trajectories of the Lorenz 96 system (17) for different forces $F$.

force of $F = 0.9$ as well as a trajectory for a large external force of $F = 8$. We can see that while for $F = 0.9$ the system does not exhibit chaotic behavior, the dynamics for $F = 8$ is already highly chaotic.

Our task consists of predicting the 25-th next state of a trajectory of the system (17). We provide 128 trajectories of length 2000 for each of the training, validation and test sets. The trajectories are generated by numerically solving the system (17) and evaluating it at 2000 equidistantly distributed discrete time points with distance 0.01. The initial value for each trajectory is chosen uniform at random on $[F - 1/2, F + 1/2]^5$ around the equilibrium point $(F, \ldots, F)$ of the system (17).

Since LSTMs are known to be able to produce chaotic dynamics, even in the autonomous (zero-entry) case (Laurent & von Brecht, 2017), we expect them to perform significantly better than coRNN if the underlying system exhibits strong chaotic behavior. Table 6 shows the normalized root mean square error (NRMSE) (RMSE divided by the root mean square of the target trajectory) on the test set for coRNN and LSTM. We can see that indeed for the non-chaotic case of using an external force of $F = 0.9$ LSTM and coRNN perform similarly. However, when the dynamics get chaotic (in this case using an external force of $F = 8$), the LSTM clearly outperforms coRNN.

Table 6: Test NRMSE on the Lorenz 96 system (17) for coRNN and LSTM.

| Model | $F = 0.9$ | $F = 8$ | # units | # params |
|-------|-----------|---------|---------|----------|
| LSTM  | $2.0 \times 10^{-2}$ | $6.8 \times 10^{-2}$ | 44 | 9k |
| coRNN | $2.0 \times 10^{-2}$ | $9.8 \times 10^{-2}$ | 64 | 9k |

## B    TRAINING DETAILS

The IMDB task was conducted on an NVIDIA GeForce GTX 1080 Ti GPU, while all other experiments were run on a Intel Xeon E3-1585Lv5 CPU. The weights and biases of coRNN are randomly initialized according to $\mathcal{U}(-\frac{1}{\sqrt{n_{in}}}, \frac{1}{\sqrt{n_{in}}})$, where $n_{in}$ denotes the input dimension of each affine transformation. Instead of treating the parameters $\Delta t, \gamma$ and $\epsilon$ as fixed hyperparameters, we can also treat them as trainable network parameters by constraining $\Delta t$ to $[0, 1]$ by using a sigmoidal activation function and $\epsilon, \gamma > 0$ by the use of ReLU for instance. However, in this case no major difference in performance is obtained. The hyperparameters are optimized with a random search algorithm, where the results of the best performing coRNN (based on the validation set) are reported. The ranges of the hyperparameters for the random search algorithm are provided in Table 7. Table 8 shows the rounded hyperparameters of the best performing coRNN architecture resulting from the random search algorithm for each learning task. We used 100 training epochs for sMNIST, psMNIST and noise padded CIFAR-10 with additional 20 epochs in which the learning rate was reduced by a factor of 10. Additionally, we used 100 epochs for the IMDB task and 250 epochs for the HAR-2 task.

Table 7: Setting for the hyperparameter optimization of coRNN. Intervals denote ranges of the corresponding hyperparameter for the grid search algorithm, while fixed numbers mean that no hyperparameter optimization was done in this case.

| task | learning rate | batch size | $\Delta t$ | $\gamma$ | $\epsilon$ |
|---|---|---|---|---|---|
| Adding | $2 \times 10^{-2}$ | 50 | $[10^{-2}, 10^{-1}]$ | $[1, 100]$ | $[1, 100]$ |
| sMNIST ($n_{hid} = 128$) | $[10^{-4}, 10^{-1}]$ | 120 | $[10^{-2}, 10^{-1}]$ | $[10^{-1}, 10]$ | $[10^{-1}, 10]$ |
| sMNIST ($n_{hid} = 256$) | $[10^{-4}, 10^{-1}]$ | 120 | $[10^{-2}, 10^{-1}]$ | $[10^{-1}, 10]$ | $[10^{-1}, 10]$ |
| psMNIST ($n_{hid} = 128$) | $[10^{-4}, 10^{-1}]$ | 120 | $[10^{-2}, 10^{-1}]$ | $[10^{-1}, 10]$ | $[10^{-1}, 10]$ |
| psMNIST ($n_{hid} = 256$) | $[10^{-4}, 10^{-1}]$ | 120 | $[10^{-2}, 10^{-1}]$ | $[10^{-1}, 10]$ | $[10^{-1}, 10]$ |
| Noise padded CIFAR-10 | $[10^{-4}, 10^{-1}]$ | 100 | $[10^{-2}, 10^{-1}]$ | $[1, 100]$ | $[1, 100]$ |
| HAR-2 | $[10^{-4}, 10^{-1}]$ | 64 | $[10^{-2}, 10^{-1}]$ | $[10^{-1}, 10]$ | $[10^{-1}, 10]$ |
| IMDB | $[10^{-4}, 10^{-1}]$ | 64 | $[10^{-2}, 10^{-1}]$ | $[10^{-1}, 10]$ | $[10^{-1}, 10]$ |

Table 8: Rounded hyperparameters of the best performing coRNN architecture.

| task | learning rate | batch size | $\Delta t$ | $\gamma$ | $\epsilon$ |
|---|---|---|---|---|---|
| Adding ($T = 5000$) | $2 \times 10^{-2}$ | 50 | $1.6 \times 10^{-2}$ | 94.5 | 9.5 |
| sMNIST ($n_{hid} = 128$) | $3.5 \times 10^{-3}$ | 120 | $5.3 \times 10^{-2}$ | 1.7 | 4 |
| sMNIST ($n_{hid} = 256$) | $2.1 \times 10^{-3}$ | 120 | $4.2 \times 10^{-2}$ | 2.7 | 4.7 |
| psMNIST ($n_{hid} = 128$) | $3.7 \times 10^{-3}$ | 120 | $8.3 \times 10^{-2}$ | $1.3 \times 10^{-1}$ | 4.1 |
| psMNIST ($n_{hid} = 256$) | $5.4 \times 10^{-3}$ | 120 | $7.6 \times 10^{-2}$ | $4 \times 10^{-1}$ | 8.0 |
| Noise padded CIFAR-10 | $7.5 \times 10^{-3}$ | 100 | $3.4 \times 10^{-2}$ | 1.3 | 12.7 |
| HAR-2 | $1.7 \times 10^{-2}$ | 64 | $10^{-1}$ | $2 \times 10^{-1}$ | 6.4 |
| IMDB | $6.0 \times 10^{-4}$ | 64 | $5.4 \times 10^{-2}$ | 4.9 | 4.8 |

## C    HEURISTICS OF NETWORK FUNCTION

At the level of a single neuron, the dynamics of the RNN is relatively straightforward. We start with the scalar case, i.e. $m = d = 1$ and illustrate different hidden states $\mathbf{y}$ as a function of time, for different input signals, in Fig. 6. In this figure, we consider two different input signals, one oscillatory signal given by $\mathbf{u}(t) = \cos(4t)$ and another is a combination of step functions. First, we plot the solution $\mathbf{y}(t)$ of (1), with the parameters $\mathbf{V}, \mathbf{b}, \mathbf{W}, \mathcal{W}, \epsilon = 0$ and $\gamma = 1$. This simply corresponds to the case of a simple harmonic oscillator (SHO) and the solution is described by a sine wave with the natural frequency of the oscillator. Next, we introduce forcing by the input signal by setting $\mathbf{V} = 1$ and the activation function is the identity $\sigma(x) = x$, leading to a forced damped oscillator (FDO). As seen from Fig. 6, in the case of an oscillatory signal, this leads to a very minor change over the SHO,

whereas for the step function, the change is only in the amplitude of the wave. Next, we add damping by setting $\epsilon = 0.25$ and see that the resulting forced damped oscillator (FDO), merely damps the amplitude of the waves, without changing their frequency. Then, we consider the case of *controlled oscillator* (CFDO) by setting $\mathbf{W} = -2, \mathbf{V} = 2, \mathbf{b} = 0.25, \mathcal{W} = 0.75$. As seen from Fig. 6, this leads to a significant change in the wave form in both cases. For the oscillatory input, the output is now a superposition of many different forms, with different amplitudes and frequencies (phases) whereas for the step function input, the phase is shifted. Already, we can see that for a linear controlled oscillator, the output can be very complicated with the superposition of different waves. This holds true when the activation function is set to $\sigma(x) = \tanh(x)$ (which is our proposed coRNN). For both inputs, the output is a modulated version of the one generated by CFDO, expressed as a superposition of waves. On the other hand, we also plot the solution with a Duffing type oscillator (DUFF) by setting the activation function as,

$$\sigma(x) = x - \frac{x^3}{3}. \tag{18}$$

In this case, the solution is very different from the CFDO and coRNN solutions and is heavily damped (either in the output or its derivative). On the other hand, given the chaotic nature of the dynamical system in this case, a slight change in the parameters led to the output blowing up. Thus, a bounded nonlinearity seems essential in this context.

Coupling neurons together further accentuates this generation of superpositions of different wave-forms, as seen even with the simplest case of a network with two neurons, shown in Fig. 6 (Bottom row). For this figure, we consider two neurons, i.e $m = 2$ and two different network topologies. For the first, we only allow the first neuron to influence the second one and not vice versa. This is enforced with the weight matrices,

$$\mathbf{W} = \begin{bmatrix} -2 & 0 \\ 3 & -2 \end{bmatrix}, \quad \mathcal{W} = \begin{bmatrix} 0.75 & 0 \\ -1 & 0.75 \end{bmatrix}.$$

We also set $\mathbf{V} = [2, 2]^\top, \mathbf{b} = [0.25, 0.25]^\top$. Note that in this case (we name as ORD (for ordered connections)), the output of the first neuron should be exactly the same as in the uncoupled (UC) case, whereas there is a distinct change in the output of the second neuron and we see that the first neuron has modulated a sharp change in the resulting output wave form. It is well illustrated by the emergence of an approximation to the step function (Bottom Right of Fig. 6), even though the input signal is oscillatory.

Next, we consider the case of fully connected (FC) neurons by setting the weight matrices as,

$$\mathbf{W} = \begin{bmatrix} -2 & 1 \\ 3 & -2 \end{bmatrix}, \quad \mathcal{W} = \begin{bmatrix} 0.75 & 0.3 \\ -1 & 0.75 \end{bmatrix}.$$

The resulting outputs for the first neuron are now slightly different from the uncoupled case. On the the other hand, the approximation of step function output for the second neuron is further accentuated.

Even these simple examples illustrate the functioning of a network of controlled oscillators well. The input signal is converted into a superposition of waves with different frequencies and amplitudes, with these quantities being controlled by the weights and biases in (1). Thus, very complicated outputs can be generated by modulating the number, frequencies and amplitudes of the waves. In practice, a network of a large number of neurons is used and can lead to extremely rich global dynamics, along the lines of emergence of synchronization or bistable heterogeneous behavior seen in systems of idealized oscillators and explained by their mean field limit, see H. Sakaguchi & Kuramoto (1987); Winfree (1967); Strogatz (2001). Thus, we argue that the ability of the network of (forced, driven) oscillators to access a very rich set of output states can lead to high expressivity of the system. The training process selects the weights that modulate frequencies, phases and amplitudes of individual neurons and their interaction to guide the system to its target output.

# D  BOUNDS ON THE DYNAMICS OF THE ORDINARY DIFFERENTIAL EQUATION (1)

In this section, we present bounds that show how the continuous time dynamics of the ordinary differential equation (2), modeling non-linear damped and forced networks of oscillators, is constrained. We start with the following estimate on the *energy* of the solutions of the system (2).

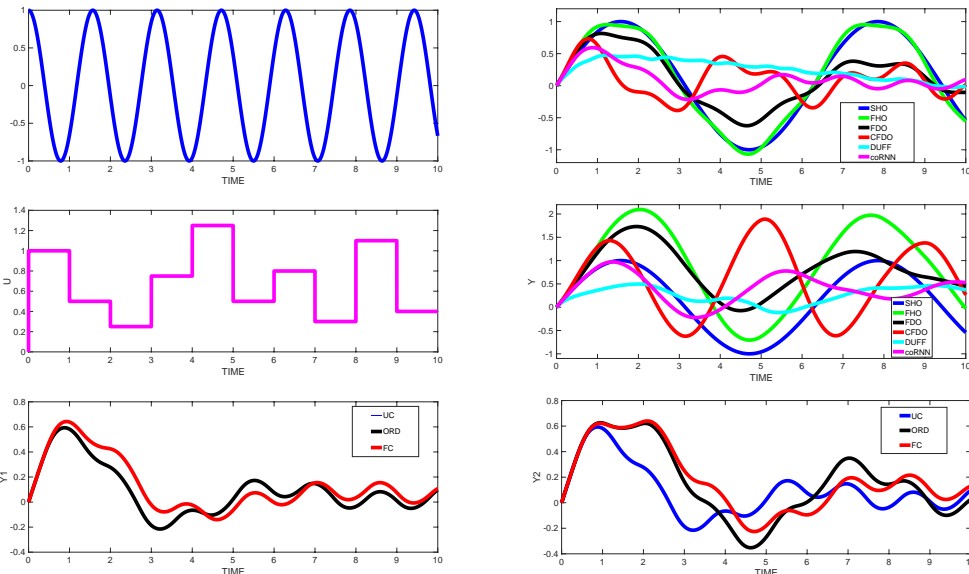

Figure 6: Illustration of the hidden state $\mathbf{y}$ of coRNN (3) with a scalar input signal $\mathbf{u}$ (Top, Middle, Left) with one neuron with state $\mathbf{y}$ (Top and Middle, Right) and two neurons with states $\mathbf{y}_1$ (Bottom left), and $\mathbf{y}_2$ (Bottom right), corresponding to scalar input signal, shown in Top Left. Legend is SHO (simple harmonic oscillator), FHO (forced oscillator), FDO (forced and damped oscillator), CFDO (controlled forced and damped oscillator), DUFF (Duffing type) UC (Uncoupled), Ord (ordered coupling) and FC (fully coupled). Legend explained in the text.

**Proposition D.1** *Let* $\mathbf{y}(t), \mathbf{z}(t)$ *be the solutions of the ODE system* (2) *at any time* $t \in [0, T]$ *and assume that the damping parameter* $\epsilon \geq \frac{1}{2}$ *and the initial data for* (2) *is given by,*

$$\mathbf{y}(0) = \mathbf{z}(0) \equiv 0.$$

*Then, the solutions are bounded as,*

$$\mathbf{y}(t)^\top \mathbf{y}(t) \leq \frac{mt}{\gamma}, \quad \mathbf{z}(t)^\top \mathbf{z}(t) \leq mt, \quad \forall t \in (0, T]. \tag{19}$$

To prove this proposition, we multiply the first equation in (2) with $y(t)^\top$ and the second equation in (2) with $\frac{1}{\gamma} z(t)^\top$ to obtain,

$$\frac{d}{dt} \left( \frac{\mathbf{y}(t)^\top \mathbf{y}(t)}{2} + \frac{\mathbf{z}(t)^\top \mathbf{z}(t)}{2\gamma} \right) = \frac{\mathbf{z}(t)^\top \sigma(\mathbf{A}(t))}{\gamma} - \frac{\epsilon}{\gamma} \mathbf{z}(t)^\top \mathbf{z}(t), \tag{20}$$

with

$$\mathbf{A}(t) = \mathbf{W}\mathbf{y}(t) + \mathcal{W}\mathbf{z}(t) + \mathbf{V}\mathbf{u}(t) + \mathbf{b}.$$

Using the elementary Cauchy's inequality repeatedly in (20) results in,

$$\frac{d}{dt} \left( \frac{\mathbf{y}(t)^\top \mathbf{y}(t)}{2} + \frac{\mathbf{z}(t)^\top \mathbf{z}(t)}{2\gamma} \right) \leq \frac{\sigma(A)^\top \sigma(A)}{2\gamma} + \frac{1}{\gamma} \left( \frac{1}{2} - \epsilon \right) \mathbf{z}^\top \mathbf{z}$$

$$\leq \frac{m}{2\gamma} \quad \left( \text{as } |\sigma| \leq 1 \text{ and } \epsilon \geq \frac{1}{2} \right).$$

Integrating the above inequality over the time interval $[0, t]$ and using the fact that the initial data are $\mathbf{y}(0) = \mathbf{z}(0) \equiv 0$, we obtain the bounds (19).

The above proposition and estimate (19) clearly demonstrate that the dynamics of the network of coupled non-linear oscillators (1) is bounded. The fact that the nonlinear activation function $\sigma = \tanh$ is uniformly bounded in its arguments played a crucial role in deriving the energy bound (19). A straightforward adaptation of this argument leads to the following proposition about the sensitivity of the system to inputs,

**Proposition D.2** *Let* $\mathbf{y}(t), \mathbf{z}(t)$ *be the solutions of the ODE system* (2) *with respect to the input signal* $\mathbf{u}(t)$. *Let* $\bar{\mathbf{y}}(t), \bar{\mathbf{z}}(t)$ *be the solutions of the ODE system* (2), *but with respect to the input signal* $\bar{\mathbf{u}}(t)$. *Assume that the damping parameter* $\epsilon \geq \frac{1}{2}$ *and the initial data are given by,*

$$\mathbf{y}(0) = \mathbf{z}(0) = \bar{\mathbf{y}}(0) = \bar{\mathbf{z}}(0) \equiv 0.$$

*Then we have the following bound,*

$$(\mathbf{y}(t) - \bar{\mathbf{y}}(t))^\top (\mathbf{y}(t) - \bar{\mathbf{y}}(t)) \leq \frac{4mt}{\gamma}, \quad (\mathbf{z}(t) - \bar{\mathbf{z}}(t))^\top (\mathbf{z}(t) - \bar{\mathbf{z}}(t)) \leq 4mt, \quad \forall t \in (0, T].$$
$$(21)$$

Thus from the bound (21), there can be *atmost* linear separation (in time) with respect to the trajectories of the ODE (2) for different input signals. Hence, chaotic behavior, which is characterized by the (super-)exponential separation of trajectories is ruled out by the structure of the ODE system (2). Note that this property of the ODE system was primarily a result of the uniform boundedness of the activation function $\sigma$. Using a different activation function such as ReLU might enable to obtain an exponential separation of trajectories that is a prerequisite for a chaotic dynamical system.

### D.1 GRADIENT DYNAMICS FOR THE ODE SYSTEM (2)

Let $\theta$ denote the $i, j$-th entry of the Weight matrices $\mathbf{W}, \mathcal{W}, \mathbf{V}$ or the $i$-th entry of the bias vector $\mathbf{b}$. We are interested in finding out how the gradients of the hidden state $\mathbf{y}$ (and the auxiliary hidden state $\mathbf{z}$) with respect to parameter $\theta$, vary with time. Note that these gradients are precisely the objects of interest in the training of an RNN, based on a discretization of the ODE system (2). To this end, we differentiate (2) with respect to the parameter $\theta$ and denote

$$\mathbf{y}_\theta(t) = \frac{\partial \mathbf{y}}{\partial \theta}(t), \ \mathbf{z}_\theta(t) = \frac{\partial \mathbf{z}}{\partial \theta}(t),$$

to obtain,

$$\begin{aligned} \mathbf{y}'_\theta &= \mathbf{z}_\theta, \\ \mathbf{z}'_\theta &= \mathrm{diag}(\sigma'(\mathbf{A})) \left[ \mathbf{W}\mathbf{y}_\theta + \mathcal{W}\mathbf{z}_\theta \right] + \mathbf{Z}^{i,j}_{m,\bar{m}}(\mathbf{A})\rho - \gamma\mathbf{y}_\theta - \epsilon\mathbf{z}_\theta. \end{aligned}$$
$$(22)$$

As introduced before, $\mathbf{Z}^{i,j}_{m,\bar{m}}(\mathbf{A}) \in \mathbb{R}^{m \times \bar{m}}$ is a matrix with all elements are zero except for the $(i, j)$-th entry which is set to $\sigma'(\mathbf{A}(t))_i$, i.e. the $i$-th entry of $\sigma'(\mathbf{A})$, and we have,

$$\begin{aligned} \rho &= \mathbf{y}, & \bar{m} &= m, & \text{if } \theta = \mathbf{W}_{i,j}, \\ \rho &= \mathbf{z}, & \bar{m} &= m, & \text{if } \theta = \mathcal{W}_{i,j}, \\ \rho &= \mathbf{u}, & \bar{m} &= d, & \text{if } \theta = \mathbf{V}_{i,j}, \\ \rho &= \mathbf{1}, & \bar{m} &= 1, & \text{if } \theta = \mathbf{b}_i. \end{aligned}$$

We see from (22) that the ODEs governing the gradients with respect to the parameter $\theta$ also represent a system of oscillators but with additional coupling and forcing terms, proportional to the hidden states $\mathbf{y}, \mathbf{z}$ or input signal $\mathbf{u}$. As we have already proved with estimate (19) that the hidden states are always bounded and the input signal is assumed to be bounded, it is natural to expect that the gradients of the states with respect to $\theta$ are also bounded. We make this statement explicit in the following proposition, which for simplicity of exposition, we consider the case of $\theta = \mathbf{W}_{i,j}$, as the other values of $\theta$ are very similar in their behavior.

**Proposition D.3** *Let* $\theta = \mathbf{W}_{i,j}$ *and* $\mathbf{y}, \mathbf{z}$ *be the solutions of the ODE system* (2). *Assume that the weights and the damping parameter satisfy,*

$$\|\mathbf{W}\|_\infty + \|\mathcal{W}\|_\infty \leq \epsilon,$$

*then we have the following bounds on the gradients,*

$$\mathbf{y}_\theta(t)^\top \mathbf{y}_\theta(t) + \frac{1}{\gamma} \left( \mathbf{z}_\theta(t)^\top \mathbf{z}_\theta(t) \right) \leq \left[ \mathbf{y}_\theta(0)^\top \mathbf{y}_\theta(0) + \frac{1}{\gamma} \left( \mathbf{z}_\theta(0)^\top \mathbf{z}_\theta(0) \right) \right] e^{Ct} + \frac{mt^2}{2\gamma^2}, \quad t \in (0, T],$$

$$C = \max \left\{ \frac{\|\mathbf{W}\|_1}{\gamma}, 1 + \|\mathcal{W}\|_1 \right\}.$$
$$(23)$$

The proof of this proposition follows exactly along the same lines as the proof of proposition D.1 and we skip the details, while noting the crucial role played by the energy bound (19).

We remark that the bound (23) indicates that as long as the initial gradients with respect to $\theta$ are bounded and the weights are controlled by the damping parameter, the hidden state gradients remain bounded in time.

## E   SUPPLEMENT TO THE RIGOROUS ANALYSIS OF CORNN

In this section, we supplement the section on the rigorous analysis of the proposed RNN (4). We start with

### E.1   PROOF OF PROPOSITION 3.1

We multiply $(\mathbf{y}_{n-1}^\top, \mathbf{z}_n^\top)$ to (3) and use the elementary identities,

$$\mathbf{a}^\top(\mathbf{a}-\mathbf{b}) = \frac{\mathbf{a}^\top\mathbf{a}}{2} - \frac{\mathbf{b}^\top\mathbf{b}}{2} + \frac{1}{2}(\mathbf{a}-\mathbf{b})^\top(\mathbf{a}-\mathbf{b}), \quad \mathbf{b}^\top(\mathbf{a}-\mathbf{b}) = \frac{\mathbf{a}^\top\mathbf{a}}{2} - \frac{\mathbf{b}^\top\mathbf{b}}{2} - \frac{1}{2}(\mathbf{a}-\mathbf{b})^\top(\mathbf{a}-\mathbf{b}),$$

to obtain the following,

$$
\begin{aligned}
\frac{\mathbf{y}_n^\top\mathbf{y}_n + \mathbf{z}_n^\top\mathbf{z}_n}{2} &= \frac{\mathbf{y}_{n-1}^\top\mathbf{y}_{n-1} + \mathbf{z}_{n-1}^\top\mathbf{z}_{n-1}}{2} + \frac{(\mathbf{y}_n - \mathbf{y}_{n-1})^\top(\mathbf{y}_n - \mathbf{y}_{n-1})}{2} \\
&\quad - \frac{(\mathbf{z}_n - \mathbf{z}_{n-1})^\top(\mathbf{z}_n - \mathbf{z}_{n-1})}{2} + \Delta t\mathbf{z}_n^\top\sigma(\mathbf{A}_{n-1}) - \Delta t\mathbf{z}_n^\top\mathbf{z}_n \\
&\leq \frac{\mathbf{y}_{n-1}^\top\mathbf{y}_{n-1} + \mathbf{z}_{n-1}^\top\mathbf{z}_{n-1}}{2} + \Delta t\left(1/2 + \Delta t/2 - 1\right)\mathbf{z}_n^\top\mathbf{z}_n + \frac{\Delta t}{2}\sigma^\top(\mathbf{A}_{n-1})\sigma(\mathbf{A}_{n-1}) \\
&\leq \frac{\mathbf{y}_{n-1}^\top\mathbf{y}_{n-1} + \mathbf{z}_{n-1}^\top\mathbf{z}_{n-1}}{2} + \frac{m\Delta t}{2} \quad \text{as } \sigma^2 \leq 1 \text{ and } \epsilon > \Delta t << 1.
\end{aligned}
$$

Iterating the above inequality $n$ times leads to the energy bound,

$$\mathbf{y}_n^\top\mathbf{y}_n + \mathbf{z}_n^\top\mathbf{z}_n \leq \mathbf{y}_0^\top\mathbf{y}_0 + \mathbf{z}_0^\top\mathbf{z}_0 + nm\Delta t = mt_n, \tag{24}$$

as $\mathbf{y}_0 = \mathbf{z}_0 = 0$.

### E.2   SENSITIVITY TO INPUTS

Next, we examine how changes in the input signal $\mathbf{u}$ affect the dynamics. We have the following proposition:

**Proposition E.1** *Let* $\mathbf{y}_n, \mathbf{z}_n$ *be the hidden states of the trained RNN* (4) *with respect to the input* $\mathbf{u} = \{\mathbf{u}_n\}_{n=1}^N$ *and let* $\overline{\mathbf{y}}_n, \overline{\mathbf{z}}_n$ *be the hidden states of the same RNN* (4)*, but with respect to the input* $\overline{\mathbf{u}} = \{\overline{\mathbf{u}}_n\}_{n=1}^N$*, then the differences in the hidden states are bounded by,*

$$\left(\mathbf{y}_n - \overline{\mathbf{y}}_n\right)^\top\left(\mathbf{y}_n - \overline{\mathbf{y}}_n\right) + \left(\mathbf{z}_n - \overline{\mathbf{z}}_n\right)^\top\left(\mathbf{z}_n - \overline{\mathbf{z}}_n\right) \leq 4mt_n. \tag{25}$$

The proof of this proposition is completely analogous to the proof of proposition 3.1, we subtract

$$
\begin{aligned}
\overline{\mathbf{y}}_n &= \overline{\mathbf{y}}_{n-1} + \Delta t\overline{\mathbf{z}}_n, \\
\overline{\mathbf{z}}_n &= \frac{\overline{\mathbf{z}}_{n-1}}{1+\Delta t} + \frac{\Delta t}{1+\Delta t}\sigma(\overline{\mathbf{A}}_{n-1}) - \frac{\Delta t}{1+\Delta t}\overline{\mathbf{y}}_{n-1}, \quad \overline{\mathbf{A}}_{n-1} := \mathbf{W}\overline{\mathbf{y}}_{n-1} + \mathcal{W}\overline{\mathbf{z}}_{n-1} + \mathbf{V}\overline{\mathbf{u}}_n + \mathbf{b}.
\end{aligned}
\tag{26}
$$

from (4) and multiply $\left((\mathbf{y}_n - \overline{\mathbf{y}}_n)^\top, (\mathbf{z}_n - \overline{\mathbf{z}}_n)^\top\right)$ to the difference. The estimate (25) follows identically to the proof of (5) (presented above) by realizing that $\sigma(\mathbf{A}_{n-1}) - \sigma(\overline{\mathbf{A}}_{n-1}) \leq 2$.

Note that the bound (25) ensures that the hidden states can only separate linearly in time for changes in the input. Thus, chaotic behavior, such as for Duffing type oscillators, characterized by at least exponential separation of trajectories, is ruled out for this proposed RNN, showing that it is stable with respect to changes in the input. This is largely on account of the fact that the activation function $\sigma$ in (3) is globally bounded.

### E.3 PROOF OF PROPOSITION 3.2

From (6), we readily calculate that,

$$\frac{\partial \mathcal{E}_n}{\partial \mathbf{X}_n} = [\mathbf{y}_n - \bar{\mathbf{y}}_n, 0].\tag{27}$$

Similarly from (3), we calculate,

$$\frac{\partial^+ \mathbf{X}_k}{\partial \theta} = \begin{cases} \left[\left(\frac{\Delta t^2}{1+\Delta t}\mathbf{Z}_{m,m}^{i,j}(\mathbf{A}_{k-1})\mathbf{y}_{k-1}\right)^\top, \left(\frac{\Delta t}{1+\Delta t}\mathbf{Z}_{m,m}^{i,j}(\mathbf{A}_{k-1})\mathbf{y}_{k-1}\right)^\top\right]^\top & \text{if} \quad \theta = (i,j)-\text{th entry of } \mathbf{W}, \\[2mm] \left[\left(\frac{\Delta t^2}{1+\Delta t}\mathbf{Z}_{m,m}^{i,j}(\mathbf{A}_{k-1})\mathbf{z}_{k-1}\right)^\top, \left(\frac{\Delta t}{1+\Delta t}\mathbf{Z}_{m,m}^{i,j}(\mathbf{A}_{k-1})\mathbf{z}_{k-1}\right)^\top\right]^\top & \text{if} \quad \theta = (i,j)-\text{th entry of } \mathcal{W}, \\[2mm] \left[\left(\frac{\Delta t^2}{1+\Delta t}\mathbf{Z}_{m,d}^{i,j}(\mathbf{A}_{k-1})\mathbf{u}_k\right)^\top, \left(\frac{\Delta t}{1+\Delta t}\mathbf{Z}_{m,d}^{i,j}(\mathbf{A}_{k-1})\mathbf{u}_k\right)^\top\right]^\top & \text{if} \quad \theta = (i,j)-\text{th entry of } \mathbf{V}, \\[2mm] \left[\left(\frac{\Delta t^2}{1+\Delta t}\mathbf{Z}_{m,1}^{i,1}(\mathbf{A}_{k-1})\right)^\top, \left(\frac{\Delta t}{1+\Delta t}\mathbf{Z}_{m,1}^{i,1}(\mathbf{A}_{k-1})\right)^\top\right]^\top & \text{if} \quad \theta = i-\text{th entry of } \mathbf{b}, \end{cases}\tag{28}$$

where $\mathbf{Z}_{m,\bar{m}}^{i,j}(\mathbf{A}_{k-1}) \in \mathbb{R}^{m \times \bar{m}}$ is a matrix with all elements are zero except for the $(i,j)$-th entry which is set to $\sigma'(\mathbf{A}_{k-1})_i$, i.e. the $i$-th entry of $\sigma'(\mathbf{A}_{k-1})$. We easily see that $\|\mathbf{Z}_{m,\bar{m}}^{i,j}(\mathbf{A}_{k-1})\|_\infty \leq 1$ for all $i, j, m, \bar{m}$ and all choices of $\mathbf{A}_{k-1}$.

Now, using definitions of matrix and vector norms and applying (14) in (10), together with (27) and (28), we obtain the following estimate on the norm:

$$\left|\frac{\partial \mathcal{E}_n^{(k)}}{\partial \theta}\right| \leq \begin{cases} (\|\mathbf{y}_n\|_\infty + \|\bar{\mathbf{y}}_n\|_\infty)(1 + 3(n-k)\Delta t^r)\delta\Delta t\|\mathbf{y}_{k-1}\|_\infty, & \text{if} \quad \theta \text{ is entry of } \mathbf{W}, \\ (\|\mathbf{y}_n\|_\infty + \|\bar{\mathbf{y}}_n\|_\infty)(1 + 3(n-k)\Delta t^r)\delta\Delta t\|\mathbf{z}_{k-1}\|_\infty, & \text{if} \quad \theta \text{ is entry of } \mathcal{W}, \\ (\|\mathbf{y}_n\|_\infty + \|\bar{\mathbf{y}}_n\|_\infty)(1 + 3(n-k)\Delta t^r)\delta\Delta t\|\mathbf{u}_k\|_\infty, & \text{if} \quad \theta \text{ is entry of } \mathbf{V}, \\ (\|\mathbf{y}_n\|_\infty + \|\bar{\mathbf{y}}_n\|_\infty)(1 + 3(n-k)\Delta t^r)\delta\Delta t, & \text{if} \quad \theta \text{ is entry of } \mathbf{b}. \end{cases}\tag{29}$$

We will estimate the above term, just for the case of $\theta$ is an entry of $\mathbf{W}$, the rest of the terms are very similar to estimate.

For simplicity of notation, we let $k - 1 \approx k$ and aim to estimate the term,

$$\left|\frac{\partial \mathcal{E}_n^{(k)}}{\partial \theta}\right| \leq \|\mathbf{y}_n\|_\infty \|\mathbf{y}_k\|_\infty (1 + 3(n-k)\Delta t^r)\delta\Delta t + \|\bar{\mathbf{y}}_n\|_\infty \|\mathbf{y}_k\|_\infty (1 + 3(n-k)\Delta t^r)\delta\Delta t$$

$$\leq m\sqrt{nk}\Delta t(1 + 3(n-k)\Delta t^r)\delta\Delta t + \|\bar{\mathbf{y}}_n\|_\infty \sqrt{mk}\sqrt{\Delta t}(1 + 3(n-k)\Delta t^r)\delta\Delta t \quad \text{(by (5))}$$

$$\leq m\sqrt{nk}\delta\Delta t^2 + 3m\sqrt{nk}(n-k)\delta\Delta t^{r+2} + \|\bar{\mathbf{y}}_n\|_\infty \sqrt{mk}\sqrt{\Delta t}(1 + 3(n-k)\Delta t^r)\delta\Delta t.\tag{30}$$

To further analyze the above estimate, we recall that $n\Delta t = t_n \leq 1$ and consider two different regimes. Let us start by considering *short-term dependencies* by letting $k \approx n$, i.e $n - k = c$ with constant $c \sim \mathcal{O}(1)$, independent of $n, k$. In this case, a straightforward application of the above assumptions in the bound (30) yields,

$$\left|\frac{\partial \mathcal{E}_n^{(k)}}{\partial \theta}\right| \leq m\sqrt{nk}\delta\Delta t^2 + 3m\sqrt{nk}(n-k)\delta\Delta t^{r+2} + \|\bar{\mathbf{y}}_n\|_\infty \sqrt{m}\sqrt{t_n}\delta\Delta t + \|\bar{\mathbf{y}}_n\|_\infty \sqrt{m}\sqrt{t_n}c\delta\Delta t^{r+1}$$

$$\leq mt_n\delta\Delta t + mct_n\delta\Delta t^{r+1} + \|\bar{\mathbf{y}}_n\|_\infty \sqrt{m}\sqrt{t_n}\delta\Delta t + \|\bar{\mathbf{y}}_n\|_\infty \sqrt{m}\sqrt{t_n}c\delta\Delta t^{r+1}$$
$$\leq t_n m\delta\Delta t + \|\bar{\mathbf{y}}_n\|_\infty \sqrt{m}\sqrt{t_n}\delta\Delta t \quad \text{(for } \Delta t << 1 \text{ as } r \geq 1/2)$$
$$\leq m\delta\Delta t + \|\bar{\mathbf{y}}_n\|_\infty \sqrt{m}\delta\Delta t.$$
$$\tag{31}$$

Next, we consider *long-term dependencies* by setting $k << n$ and estimating,

$$\left|\frac{\partial \mathcal{E}_n^{(k)}}{\partial \theta}\right| \leq m\sqrt{nk}\delta\Delta t^2 + 3m\sqrt{nk}(n-k)\delta\Delta t^{r+2} + \|\bar{\mathbf{y}}_n\|_\infty\sqrt{m}\delta\Delta t^{\frac{3}{2}} + 3\|\bar{\mathbf{y}}_n\|_\infty\sqrt{m}n\delta\Delta t^{r+\frac{3}{2}}$$

$$\leq m\sqrt{t_n}\delta\Delta t^{\frac{3}{2}} + 3mt_n^{\frac{3}{2}}\delta\Delta t^{r+\frac{1}{2}} + \|\bar{\mathbf{y}}_n\|_\infty\sqrt{m}\delta\Delta t^{\frac{3}{2}} + 3\|\bar{\mathbf{y}}_n\|_\infty\sqrt{m}t_n\delta\Delta t^{r+\frac{1}{2}}$$

$$\leq m\delta\Delta t^{\frac{3}{2}} + 3m\delta\Delta t^{r+\frac{1}{2}} + \|\bar{\mathbf{y}}_n\|_\infty\sqrt{m}\delta\Delta t^{\frac{3}{2}} + 3\|\bar{\mathbf{y}}_n\|_\infty\sqrt{m}\delta\Delta t^{r+\frac{1}{2}} \quad (\text{as } t_n < 1)$$

$$\leq 3m\delta\Delta t^{r+\frac{1}{2}} + 3\|\bar{\mathbf{y}}_n\|_\infty\sqrt{m}\delta\Delta t^{r+\frac{1}{2}} \quad (\text{as } r \leq 1 \text{ and } \Delta t << 1). \tag{32}$$

Thus, in all cases, we have that,

$$\left|\frac{\partial \mathcal{E}_n^{(k)}}{\partial \theta}\right| \leq 3\delta\Delta t\left(m + \sqrt{m}\|\bar{\mathbf{y}}_n\|_\infty\right) \quad (\text{as } r \geq 1/2). \tag{33}$$

Applying the above estimate in (10) allows us to bound the gradient by,

$$\left|\frac{\partial \mathcal{E}_n}{\partial \theta}\right| \leq \sum_{1 \leq k \leq n}\left|\frac{\partial \mathcal{E}_n^{(k)}}{\partial \theta}\right| \leq 3\delta t_n\left(m + \sqrt{m}\|\bar{\mathbf{y}}_n\|_\infty\right). \tag{34}$$

Therefore, the gradient of the loss function (6) can be bounded as,

$$\begin{aligned}
\left|\frac{\partial \mathcal{E}}{\partial \theta}\right| &\leq \frac{1}{N}\sum_{n=1}^{N}\left|\frac{\partial \mathcal{E}_n}{\partial \theta}\right| \\
&\leq 3\delta\left[\frac{m\Delta t}{N}\sum_{n=1}^{N}n + \frac{\sqrt{m}\Delta t}{N}\sum_{n=1}^{N}\|\bar{\mathbf{y}}_n\|_\infty n\right] \\
&\leq 3\delta\left[\frac{m\Delta t}{N}\sum_{n=1}^{N}n + \frac{\sqrt{m}\bar{Y}\Delta t}{N}\sum_{n=1}^{N}n\right] \\
&\leq \frac{3}{2}\delta(N+1)\Delta t\left(m + \bar{Y}\sqrt{m}\right) \\
&\leq \frac{3}{2}\delta(t_N + \Delta t)\left(m + \bar{Y}\sqrt{m}\right) \\
&\leq \frac{3}{2}\delta(1 + \Delta t)\left(m + \bar{Y}\sqrt{m}\right) \quad (\text{as } t_N = 1) \\
&\leq \frac{3}{2}\left(m + \bar{Y}\sqrt{m}\right),
\end{aligned} \tag{35}$$

which is the desired estimate (9).

### E.4   ON THE ASSUMPTION (8) AND TRAINING

Note that all the estimates were based on the fact that we were able to choose a time step $\Delta t$ in (3) that enforces the condition (8). For any fixed weights $\mathbf{W}, \mathcal{W}$, we can indeed choose such a value of $\epsilon$ to satisfy (8). However, we *train* the RNN to find the weights that minimize the loss function (6). Can we find a hyperparameter $\Delta t$ such that (8) is satisfied at every step of the stochastic gradient descent method for training?

To investigate this issue, we consider a simple gradient descent method of the form:

$$\theta_{\ell+1} = \theta_\ell - \zeta\frac{\partial \mathcal{E}}{\partial \theta}(\theta_\ell). \tag{36}$$

Note that $\zeta$ is the constant (non-adapted) learning rate. We assume for simplicity that $\theta_0 = 0$ (other choices lead to the addition of a constant). Then, a straightforward estimate on the weight is given by,

$$|\theta_{\ell+1}| \leq |\theta_\ell| + \zeta \left| \frac{\partial \mathcal{E}}{\partial \theta}(\theta_\ell) \right|$$

$$\leq |\theta_\ell| + \zeta \frac{3}{2} \left( m + \bar{Y}\sqrt{m} \right) \quad \text{(by (35))} \tag{37}$$

$$\leq |\theta_0| + \ell\zeta \frac{3}{2} \left( m + \bar{Y}\sqrt{m} \right) = \ell\zeta \frac{3}{2} \left( m + \bar{Y}\sqrt{m} \right).$$

In order to calculate the minimum number of steps $L$ in the gradient descent method (36) such that the condition (8) is satisfied, we set $\ell = L$ in (37) and applying it to the condition (8) leads to the straightforward estimate,

$$L \geq \frac{1}{\zeta \frac{3}{2} \left( m + \bar{Y}\sqrt{m} \right) m\Delta t^{1-r}\delta}. \tag{38}$$

Note that the parameter $\delta < 1$, while in general, the learning rate $\zeta << 1$. Thus, as long as $r \leq 1$, we see that the assumption (8) holds for a large number of steps of the gradient descent method. We remark that the above estimate (38) is a large underestimate on $L$. In the experiments presented in this article, we are able to take a very large number of training steps, while the gradients remain within a range (see Fig. 3).

### E.5   PROOF OF PROPOSITION 3.3

We start with the following decomposition of the recurrent matrices:

$$\frac{\partial \mathbf{X}_i}{\partial \mathbf{X}_{i-1}} = M_{i-1} + \Delta t \tilde{M}_{i-1},$$

$$M_{i-1} := \begin{bmatrix} \mathbf{I} & \Delta t \mathbf{C}_{i-1} \\ \mathbf{B}_{i-1} & \mathbf{C}_{i-1} \end{bmatrix}, \quad \tilde{M}_{i-1} := \begin{bmatrix} \mathbf{B}_{i-1} & \mathbf{0} \\ \mathbf{0} & \mathbf{0} \end{bmatrix},$$

with $\mathbf{B}, \mathbf{C}$ defined in (12). By the assumption (8), one can readily check that $\|\tilde{M}_{i-1}\|_\infty \leq \Delta t$, for all $k \leq i \leq n-1$.

We will use an induction argument to show the following representation formula for the product of Jacobians,

$$\frac{\partial \mathbf{X}_n}{\partial \mathbf{X}_k} = \prod_{k < i \leq n} \frac{\partial \mathbf{X}_i}{\partial \mathbf{X}_{i-1}} = \begin{bmatrix} \mathbf{I} & \Delta t \sum_{j=k}^{n-1} \prod_{i=j}^{k} \mathbf{C}_i \\ \mathbf{B}_{n-1} + \sum_{j=n-2}^{k} \left( \prod_{i=n-1}^{j+1} \mathbf{C}_i \right) \mathbf{B}_j & \prod_{i=n-1}^{k} \mathbf{C}_i \end{bmatrix} + \mathcal{O}(\Delta t). \tag{39}$$

We start by the outermost product and calculate,

$$\frac{\partial \mathbf{X}_n}{\partial \mathbf{X}_{n-1}} \frac{\partial \mathbf{X}_{n-1}}{\partial \mathbf{X}_{n-2}} = \left( M_{n-1} + \Delta t \tilde{M}_{n-1} \right) \left( M_{n-2} + \Delta t \tilde{M}_{n-2} \right)$$

$$= M_{n-1} M_{n-2} + \Delta t (\tilde{M}_{n-1} M_{n-2} + M_{n-1} \tilde{M}_{n-2}) + \mathcal{O}(\Delta t^2).$$

By direct multiplication, we obtain,

$$M_{n-1} M_{n-2} = \begin{bmatrix} \mathbf{I} & \Delta t \left( \mathbf{C}_{n-2} + \mathbf{C}_{n-1}\mathbf{C}_{n-2} \right) \\ \mathbf{B}_{n-1} + \mathbf{C}_{n-1}\mathbf{B}_{n-2} & \mathbf{C}_{n-1}\mathbf{C}_{n-2} \end{bmatrix}$$

$$+ \Delta t \begin{bmatrix} \mathbf{C}_{n-1}\mathbf{B}_{n-2} & \mathbf{0} \\ \mathbf{0} & \mathbf{B}_{n-1}\mathbf{C}_{n-2} \end{bmatrix}.$$

Using the definitions in (12) and (8), we can easily see that

$$\begin{bmatrix} \mathbf{C}_{n-1}\mathbf{B}_{n-2} & \mathbf{0} \\ \mathbf{0} & \mathbf{B}_{n-1}\mathbf{C}_{n-2} \end{bmatrix} = \mathcal{O}(\Delta t).$$

Similarly, it is easy to show that

$$\tilde{M}_{n-1} M_{n-2}, M_{n-1} \tilde{M}_{n-2} \sim \mathcal{O}(\Delta t).$$

Plugging all the above estimates yields,

$$\frac{\partial \mathbf{X}_n}{\partial \mathbf{X}_{n-1}} \frac{\partial \mathbf{X}_{n-1}}{\partial \mathbf{X}_{n-2}} = \begin{bmatrix} \mathbf{I} & \Delta t \left( \mathbf{C}_{n-2} + \mathbf{C}_{n-1} \mathbf{C}_{n-2} \right) \\ \mathbf{B}_{n-1} + \mathbf{C}_{n-1} \mathbf{B}_{n-2} & \mathbf{C}_{n-1} \mathbf{C}_{n-2} \end{bmatrix} + \mathcal{O}(\Delta t^2),$$

which is exactly the form of the leading term (39).

Iterating the above calculations $(n-k)$ times and realizing that $(n-k)\Delta t^2 \approx n\Delta t^2 = t_n \Delta t$ yields the formula (39).

Recall that we have set $\theta = \mathbf{W}_{i,j}$, for some $1 \le i,j \le m$ in proposition 3.3. Directly calculating with (27), (28) and the representation formula (39) yields the formula,

$$\frac{\partial \mathcal{E}_n^{(k)}}{\partial \theta} = \mathbf{y}_n^\top \Delta t^2 \delta \mathbf{Z}_{m,m}^{i,j}(\mathbf{A}_{k-1}) \mathbf{y}_{k-1} + \mathbf{y}_n^\top \Delta t^2 \delta \mathbf{C}^* \mathbf{Z}_{m,m}^{i,j}(\mathbf{A}_{k-1}) \mathbf{y}_{k-1} + \mathcal{O}(\Delta t^3), \qquad (40)$$

with matrix $\mathbf{C}^*$ defined as,

$$\mathbf{C}^* := \sum_{j=k}^{n-1} \prod_{i=j}^{k} \mathbf{C}_i,$$

and $\mathbf{Z}_{m,m}^{i,j}(\mathbf{A}_{k-1}) \in \mathbb{R}^{m \times m}$ is a matrix with all elements are zero except for the $(i,j)$-th entry which is set to $\sigma'(a_{k-1}^i)$, i.e. the $i$-th entry of $\sigma'(\mathbf{A}_{k-1})$.

Note that the formula (40) can be explicitly written as,

$$\frac{\partial \mathcal{E}_n^{(k)}}{\partial \theta} = \delta \Delta t^2 \sigma'(a_{k-1}^i) \mathbf{y}_n^i \mathbf{y}_{k-1}^j + \delta \Delta t^2 \sigma'(a_{k-1}^i) \sum_{\ell=1}^m \mathbf{C}_{\ell i}^* \mathbf{y}_n^\ell \mathbf{y}_{k-1}^j + \mathcal{O}(\Delta t^3), \qquad (41)$$

with $\mathbf{y}_n^j$ denoting the $j$-th element of vector $\mathbf{y}_n$, and

$$a_{k-1}^i := \sum_{\ell=1}^m \mathbf{W}_{i\ell} \mathbf{y}_{k-1}^\ell + \sum_{\ell=1}^m \mathcal{W}_{i\ell} \mathbf{z}_{k-1}^\ell. \qquad (42)$$

By the assumption (8), we can readily see that

$$\|\mathbf{W}\|_\infty, \|\mathcal{W}\|_\infty \le 1 + \Delta t.$$

Therefore by the fact that $\sigma' = sech^2$, the assumption $\mathbf{y}_k^i = \mathcal{O}(\sqrt{t_k})$ and (42), we obtain,

$$\hat{c} = sech^2(\sqrt{k\Delta t}(1 + \Delta t) \le \sigma'(a_i^{k-1}) \le 1. \qquad (43)$$

Using (43) in (41), we obtain,

$$\delta \Delta t^2 \sigma'(a_{k-1}^i) \mathbf{y}_n^i \mathbf{y}_{k-1}^j = \mathcal{O}\left(\hat{c}\delta\Delta t^{\frac{5}{2}}\right). \qquad (44)$$

Using the definition of $\mathbf{C}_i$, we can expand the product in $\mathbf{C}^*$ and neglect terms of order $\mathcal{O}(\Delta t^4)$, to obtain

$$\prod_{i=j}^k \mathbf{C}_i = (\mathcal{O}(1) + \mathcal{O}((j-k+1)\delta\Delta t^2))\mathbf{I}.$$

Summing over $j$ and using the fact that $k << n$, we obtain that

$$\mathbf{C}^* = (\mathcal{O}(n) + \mathcal{O}(\delta\Delta t^0))\mathbf{I}. \qquad (45)$$

Plugging (45) and (43) into (41) leads to,

$$\delta \Delta t^2 \sigma'(a_{k-1}^i) \sum_{\ell=1}^m \mathbf{C}_{\ell i}^* \mathbf{y}_n^\ell \mathbf{y}_{k-1}^j = \mathcal{O}\left(\hat{c}\delta\Delta t^{\frac{3}{2}}\right) + \mathcal{O}\left(\hat{c}\delta^2\Delta t^{\frac{5}{2}}\right). \qquad (46)$$

Combining (44) and (46) yields the desired estimate (16).

**Remark.** A careful examination of the above proof reveals that the constants hidden in the prefactors of the leading term $\mathcal{O}\left(\hat{c}\delta\Delta t^{\frac{3}{2}}\right)$ of (16) stem from the formula (46). Here, we have used the assumption that $\mathbf{y}_k^i = \mathcal{O}(\sqrt{t_k})$. Note that this assumption implicitly assumes that the energy bound (5) is *equidistributed* among all the elements of the vector $\mathbf{y}_k$ and results in the obfuscation of the constants in the leading term of (16). Given that the energy bound (5) is too coarse to allow for precise upper and lower bounds on each individual element of the hidden state vector $\mathbf{y}_k$, we do not see any other way of, in general, determining the distribution of energy among individual entries of the hidden state vector. Thus, assuming equidistribution seems reasonable. On the other hand, in practice, one has access to all the terms in formula (46) for each numerical experiment and if one is interested, then one can directly evaluate the precise bound on the leading term of the formula (16).

# F RIGOROUS ESTIMATES FOR THE RNN (3) WITH $\bar{n} = n - 1$ AND GENERAL VALUES OF $\epsilon, \gamma$

In this section, we will provide rigorous estimates, similar to that of propositions 3.1, E.1 and 3.2 for the version of coRNN (3) that results by setting $\bar{n} = n - 1$ in (3) leading to,

$$
\begin{aligned}
\mathbf{y}_n &= \mathbf{y}_{n-1} + \Delta t \mathbf{z}_n, \\
\mathbf{z}_n &= \mathbf{z}_{n-1} + \Delta t \sigma \left(\mathbf{W}\mathbf{y}_{n-1} + \mathcal{W}\mathbf{z}_{n-1} + \mathbf{V}\mathbf{u}_n + \mathbf{b}\right) - \Delta t \gamma \mathbf{y}_{n-1} - \Delta t \epsilon \mathbf{z}_{n-1}.
\end{aligned}
\tag{47}
$$

Note that (47) can be equivalently written as,

$$
\begin{aligned}
\mathbf{y}_n &= \mathbf{y}_{n-1} + \Delta t \mathbf{z}_n, \\
\mathbf{z}_n &= (1 - \epsilon\Delta t)\,\mathbf{z}_{n-1} + \Delta t \sigma \left(\mathbf{W}\mathbf{y}_{n-1} + \mathcal{W}\mathbf{z}_{n-1} + \mathbf{V}\mathbf{u}_n + \mathbf{b}\right) - \Delta t \gamma \mathbf{y}_{n-1}.
\end{aligned}
\tag{48}
$$

We will also consider the case of non-unit values of the control parameters $\gamma$ and $\epsilon$ below.

**Bounds on Hidden states.** We start the following bound on the hidden states of (47),

**Proposition F.1** *Let the damping parameter $\epsilon > \frac{1}{2}$ and the time step $\Delta t$ in the RNN (47) satisfy the following condition,*

$$
\Delta t < \frac{2\epsilon - 1}{\gamma + \epsilon^2}.
\tag{49}
$$

*Let $\mathbf{y}_n, \mathbf{z}_n$ be the hidden states of the RNN (47) for $1 \le n \le N$, then the hidden states satisfy the following (energy) bounds:*

$$
\mathbf{y}_n^\top \mathbf{y}_n + \frac{1}{\gamma}\mathbf{z}_n^\top \mathbf{z}_n \le \frac{mt_n}{\gamma}.
\tag{50}
$$

We set $\mathbf{A}_{n-1} = \mathbf{W}\mathbf{y}_{n-1} + \mathcal{W}\mathbf{z}_{n-1} + \mathbf{V}\mathbf{u}_{n-1} + \mathbf{b}$ and as in the proof of proposition 3.1, we multiply $(\mathbf{y}_{n-1}^\top, \frac{1}{\gamma}\mathbf{z}_n^\top)$ to (47) and use elementary identities and rearrange terms to obtain,

$$
\begin{aligned}
\frac{\mathbf{y}_n^\top \mathbf{y}_n}{2} + \frac{\mathbf{z}_n^\top \mathbf{z}_n}{2\gamma} = &\; \frac{\mathbf{y}_{n-1}^\top \mathbf{y}_{n-1}}{2} + \frac{\mathbf{z}_{n-1}^\top \mathbf{z}_{n-1}}{2\gamma} + \frac{(\mathbf{y}_n - \mathbf{y}_{n-1})^\top (\mathbf{y}_n - \mathbf{y}_{n-1})}{2} \\
&- \frac{(\mathbf{z}_n - \mathbf{z}_{n-1})^\top (\mathbf{z}_n - \mathbf{z}_{n-1})}{2\gamma} \\
&+ \frac{\Delta t}{\gamma}\mathbf{z}_n^\top \sigma(\mathbf{A}_{n-1}) - \frac{\epsilon\Delta t}{\gamma}\mathbf{z}_n^\top \mathbf{z}_n + \frac{\epsilon\Delta t}{\gamma}\mathbf{z}_n^\top (\mathbf{z}_n - \mathbf{z}_{n-1}).
\end{aligned}
$$

We use a *rescaled version* of the well-known Cauchy's inequality

$$
ab \le \frac{ca^2}{2} + \frac{b^2}{2c},
$$

for a constant $c > 0$ to be determined, to rewrite the above identity as,

$$
\begin{aligned}
\frac{\mathbf{y}_n^\top \mathbf{y}_n}{2} + \frac{\mathbf{z}_n^\top \mathbf{z}_n}{2\gamma} \le &\; \frac{\mathbf{y}_{n-1}^\top \mathbf{y}_{n-1}}{2} + \frac{\mathbf{z}_{n-1}^\top \mathbf{z}_{n-1}}{2\gamma} + \frac{(\mathbf{y}_n - \mathbf{y}_{n-1})^\top (\mathbf{y}_n - \mathbf{y}_{n-1})}{2} \\
&+ \left(\frac{\epsilon\Delta t}{2c\gamma} - \frac{1}{2\gamma}\right)(\mathbf{z}_n - \mathbf{z}_{n-1})^\top (\mathbf{z}_n - \mathbf{z}_{n-1}) + \frac{\Delta t}{2\gamma}\sigma(\mathbf{A}_{n-1})^\top \sigma(\mathbf{A}_{n-1}) \\
&+ \left(\frac{\Delta t}{2\gamma} + \frac{c\epsilon\Delta t}{2\gamma} - \frac{\epsilon\Delta t}{\gamma}\right)\mathbf{z}_n^\top \mathbf{z}_n.
\end{aligned}
$$

Using the first equation in (47), the above inequality reduces to,

$$\frac{\mathbf{y}_n^\top \mathbf{y}_n}{2} + \frac{\mathbf{z}_n^\top \mathbf{z}_n}{2\gamma} \leq \frac{\mathbf{y}_{n-1}^\top \mathbf{y}_{n-1}}{2} + \frac{\mathbf{z}_{n-1}^\top \mathbf{z}_{n-1}}{2\gamma}$$
$$+ \left( \frac{\epsilon \Delta t}{2c\gamma} - \frac{1}{2\gamma} \right) (\mathbf{z}_n - \mathbf{z}_{n-1})^\top (\mathbf{z}_n - \mathbf{z}_{n-1}) + \frac{\Delta t}{2\gamma} \sigma(\mathbf{A}_{n-1})^\top \sigma(\mathbf{A}_{n-1})$$
$$+ \left( \frac{\Delta t^2}{2} + \frac{\Delta t}{2\gamma} + \frac{c\epsilon \Delta t}{2\gamma} - \frac{\epsilon \Delta t}{\gamma} \right) \mathbf{z}_n^\top \mathbf{z}_n.$$

As long as,

$$\Delta t \leq \min \left( \frac{c}{\epsilon}, \frac{(2-c)\epsilon - 1}{\gamma} \right), \tag{51}$$

we can easily check that,

$$\frac{\mathbf{y}_n^\top \mathbf{y}_n}{2} + \frac{\mathbf{z}_n^\top \mathbf{z}_n}{2\gamma} \leq \frac{\mathbf{y}_{n-1}^\top \mathbf{y}_{n-1}}{2} + \frac{\mathbf{z}_{n-1}^\top \mathbf{z}_{n-1}}{2\gamma} + \frac{\Delta t}{2\gamma} \sigma(\mathbf{A}_{n-1})^\top \sigma(\mathbf{A}_{n-1})$$
$$\leq \frac{\mathbf{y}_{n-1}^\top \mathbf{y}_{n-1}}{2} + \frac{\mathbf{z}_{n-1}^\top \mathbf{z}_{n-1}}{2\gamma} + \frac{m\Delta t}{2\gamma} \quad (\sigma \leq 1).$$

Iterating the above bound till $n = 0$ and using the zero initial data yields the desired (50) as long as we find a $c$ such that the condition (51) is satisfied. To do so, we equalize the two terms on the right hand side of (51) to obtain,

$$c = \frac{\epsilon(2\epsilon - 1)}{\gamma + \epsilon^2}.$$

From the assumption (49) and the fact that $\epsilon > \frac{1}{2}$, we see that such a $c > 0$ always exists for any value of $\gamma > 0$ and (51) is satisfied, which completes the proof.

We remark that the same bound on the hidden states is obtained for both versions of coRNN, i.e. (3) with $\bar{n} = n$ and (47). However, the difference lies in the constraint on the time step $\Delta t$. In contrast to (49), a careful examination of the proof of proposition 3.1 reveals that the condition on the time step for the stability of (3) with $\bar{n} = n$ is given by,

$$\Delta t < \frac{2\epsilon - 1}{\gamma}, \tag{52}$$

and is clearly less stringent than the condition (51) for the stability of (47). For instance, in the prototypical case of $\gamma = \epsilon = 1$, the stability of (3) with $\bar{n} = n$ is ensured for any $\Delta t < 1$. On the other hand, the stability of (47) is ensured as long as $\Delta t < \frac{1}{2}$. However, it is essential to recall that these conditions are only sufficient to ensure stability and are by no means necessary. Thus in practice, the coRNN version (47) is found to be stable in the same range of time steps as the version (3) with $\bar{n} = n$.

**On the exploding and vanishing gradient problems for coRNN** (47)   Next, we have the following upper bound on the hidden state gradients for the version (47) of coRNN,

**Proposition F.2** *Let $\mathbf{y}_n, \mathbf{z}_n$ be the hidden states generated by the RNN (47). We assume that the damping parameter $\epsilon > \frac{1}{2}$ and the time step $\Delta t$ can be chosen such that in addition to (51) it also satisfies,*

$$\max \{ \Delta t(\gamma + \|\mathbf{W}\|_\infty), \Delta t \|\mathcal{W}\|_\infty \} = \eta \leq \tilde{C} \Delta t^r, \quad \frac{1}{2} \leq r \leq 1, \tag{53}$$

*and with the constant $\tilde{C}$ independent of the other parameters of the RNN (47). Then the gradient of the loss function $\mathcal{E}$ (6) with respect to any parameter $\theta \in \mathbf{\Theta}$ is bounded as,*

$$\left| \frac{\partial \mathcal{E}}{\partial \theta} \right| \leq \frac{3(\tilde{C}) \left( m + \bar{Y}\sqrt{m} \right)}{2\gamma}, \tag{54}$$

*with the constant $\tilde{C}$, defined in (53) and $\bar{Y} = \max_{1 \leq n \leq N} \|\bar{\mathbf{y}}_n\|_\infty$ be a bound on the underlying training data*

The proof of this proposition is completely analogous to the proof of proposition 3.2 and we omit the details here.

Note that the bound (54) enforces that hidden state gradients cannot explode for version (47) of coRNN. A similar statement for the vanishing gradient problem is inferred from the proposition below.

**Proposition F.3** *Let* $\mathbf{y}_n$ *be the hidden states generated by the RNN* (47). *Under the assumption that* $\mathbf{y}_n^i = \mathcal{O}(\sqrt{\frac{t_n}{\gamma}})$, *for all* $1 \leq i \leq m$ *and* (53), *the gradient for long-term dependencies satisfies,*

$$\frac{\partial \mathcal{E}_n^{(k)}}{\partial \theta} = \mathcal{O}\left(\frac{\hat{c}}{\gamma}\Delta t^{\frac{3}{2}}\right) + \mathcal{O}\left(\frac{\hat{c}}{\gamma}\delta(1+\delta)\Delta t^{\frac{5}{2}}\right) + \mathcal{O}(\Delta t^3), \ \ \hat{c} = sech^2\left(\sqrt{k\Delta t}(1+\Delta t)\right) \quad k << n. \tag{55}$$

The proof is a repetition of the steps of the proof of proposition 3.3, with suitable modifications for the structure of the RNN and non-unit $\epsilon, \gamma$ and we omit the tedious calculations here. Note that (55) rules out the vanishing gradient problem for the coRNN version (47).

