# OpenReview forum: "Coupled Oscillatory Recurrent Neural Network (coRNN): An accurate and (gradient) stable architecture for learning long time dependencies"
_ICLR.cc/2021/Conference — ICLR 2021 Oral_

### Official Review · AnonReviewer1 · 2020-10-21
**This paper proposes the coupled oscillatory recurrent neural networks (coRNN) for solving the exploding and vanishing gradient problem by proving precise bounds on the gradients of the hidden states.**

**Rating:** 7
**Confidence:** 3

**Review:**

Firstly, this paper conducts the rigorous analysis of the coRNN via the formula deduction to verify the bound. Then the coRNN is proved to mitigate the exploding and vanishing gradient problem and this is also validated in a series of experiments. Also, the performance of the coRNN is comparable or better compared to state-of-the-art models. This paper provides a new idea to address the exploding and vanishing gradient problem, which hinders the development of deeper neural networks tremendously. In my opinion, this coRNN model is meaningful for practical application, especially for the extension of more complicated neural networks.

Besides, for the biomedical signals with high temporal resolution (e.g., electroencephalogram, electromyogram), the coRNN model can be a good alternative in future work. Furthermore, the efficiency of the proposed model also be proved by the mathematic formulation and experiments ranging from pure synthetic tasks designed to learn long-term dependencies to more realistic tasks rigorously. Considering the whole structure of this paper, I argue that the clarity is clear and logical. Different from the recently published literature, this paper has explicit use of networks of oscillators with the underlying biological motivation, so this paper expresses the originality in some extent. To sum up, the quality of this paper is suitable for the publication in ICLR2021.

There are two main pros in this paper:
1.	The theoretical verification is clear and rigorous, readers can easily catch good understanding of the bounds this paper proves following the formula deduction. Specifically, this paper demonstrates how to avoid the exploding and vanishing gradient problem for the RNN in theory.
2.	The experiments are quite abundant, experimental results show that the coRNN can not only avoid the exploding and vanishing gradient problem, but also achieve better performance with fewer parameters compared to recent studies.

But some cons should also be noticed. Firstly, the illustration of proposed coRNN should be presented in the paper, which is more comprehensible. Secondly, the related work part, when mentioning the similar works, it will be better to describe the main differences and correction with this paper more specifically. Lastly, in the part of discussion, the practical significance of proposed coRNN should be emphasized with more words.

---

> ### Author Response · Authors · 2020-11-17
> **Reply to Reviewer 1**
>
> We start by thanking the reviewer for your appreciation of the merits of our paper and your welcome suggestions to improve it. Below, we address the very valid points raised by the reviewer.
> 1. The reviewer's concern about the write-up and illustration of coRNN is completely justified. Following the suggestions of all the reviewers, we have rewritten several parts of the paper. For instance, an intuitive explanation of the dynamics of the underlying ODE (Eqn 2) is now provided in the first two paragraphs of the section on Motivation and Background on page 2 of the revised version. In particular, we attempt to explain intuitively the functioning of a network of oscillators, first at the level of single neurons and then for a coupled system. We also explain to the reader how coRNN can access a rich set of output states. Moreover, we have tried to describe the interplay between the dynamics of the underlying ODE and the proposed coRNN which is a structure preserving discretization of this ODE. Please check the first and the last paragraphs of section 3 in this context. We hope that the revised version addresses your valid concerns about the intuitive illustration of coRNN.
> 2. We have followed your suggestion and extended the section on related work, with an emphasis on trying to differentiate our approach from other ODE inspired RNNs. In particular, we emphasize the explicit use of a network of coupled oscillators in our construction of coRNN.
> 3. We appreciate your excellent suggestion about the possible use of coRNN in the context of *high-resolution biomedical data* such as EEG, EMG etc. We believe that coRNN is very suitable for processing such data, given its construction. We have expanded the discussion to include a paragraph on the prospective use of coRNNs for processing EEG and EMG data and in this context, we would like to point out that coRNN is a promising avenue for processing seismic activity data in the geosciences. We have also added a paragraph in the discussion (on page 9) pointing out limitations of coRNN, for instance in the prediction of chaotic time series data. Thus, we hope that the current discussion provides the reader with a balanced perspective about possible uses of coRNN and addresses your legitimate concern about the lack of practical significance in the original version of the manuscript.

---

### Official Review · AnonReviewer3 · 2020-10-28
**A dynamial systems inspired CT-RNN that mitigates exploding/vanishing gradients**

**Rating:** 7
**Confidence:** 3

**Review:**

This paper proposes a new continuous-time formulation for modeling recurrent units. The particular form of the recurrent unit is motivated by a system of coupled oscillators. These systems are well studied and widely used in the physical, engineering and biological sciences. Establishing this connection has the potential to motivate interesting future works. The performance of the proposed recurrent unit is state of the art.


Reasons for my score: Overall, I vote for marginally above acceptance threshold. I like very much the proposed approach for modeling recurrent units. Further, the presented results are intriguing, and the paper is well written. However, I have some concerns (see below). I am happy to increase my score if the authors can address my concerns in the rebuttal period.


Pros:
-------
+ Second-order systems of ODEs seem to be a promising approach for modeling recurrent units, and this approach has not received much attention for this task before. Indeed, this paper impressively demonstrates that a unit motivated by a system of coupled oscillators is able to achieve state of the art performance on a range of benchmark tasks.

+ The analysis shows that the particular form of the proposed continuous-time unit mitigates the vanishing and exploding gradients problem by design, which is very appealing. The analysis is mathematically sound.

+ Code is provided!


Cons:
-------
- In Eq. (3) the authors advocate the IMEX scheme to obtain a discretization of (2). I was very curious to see how the authors implement this scheme in practice, however, the provided implementation revealed that the authors use an explicit scheme in practice. Please, comment why you chose the IMEX scheme here. Does the analysis also hold if you use an explicit discretization in (3), and if, why do you mask the fact that you are using an explicit scheme in practice. I feel, it would be relevant to discuss how you train the unit in practice.

- Section 3 is no pleasure to read. It is not clear to me what the value of the sketch of the proofs are, since the proofs are pretty standard. Instead, the space could be better used for an extended qualitative discussion of the analysis and the nice properties of the proposed recurrent unit. For instance, you can extend the discussion around proposition 3.1 and provide some context on why you want to rule out chaotic behavior (this might not be obvious for everyone); further it would be nice to see a better discussion on the effect of dampening and forcing on the performance of the recurrent unit. Also, I would like to suggest to move parts of Appendix B into the main text, since this discussion actually helps to build some intuition for the proposed unit.

- The extremely good performance on the sMNIST task is slightly surprising, since my intuition would not suggest that the particular form of the unit has the ability to substantially improve the expressivity as compared to some other recently proposed units. I used the provided code to evaluate the coRNN (N=256 and 128) on the sMNIST task and the highest accuracy that I was able to obtain (out of 8 runs on 4 different GPUs) was 99.2% on the test set. These results are still very good, but they do not match the reported results. (Note, that the code is printing out the accuracy for a smaller validation set which indicates a higher accuracy than is actually obtained on the test set.) This said, I would like to ask the authors to double check the experiments on sMNIST. (Also, I assume that the model can be trained in less time if the learning rate is decayed much earlier, e.g., around epoch 30 and a second time around epoch 60.)

- It is not clear to me how sensitive the RNN is to the particular choices of \gamma and \epsilon. It would be good to provide some form of ablation study that studies how the performance varies for different values of \gamma (and \epsilon) while keeping all other tuning parameters fixed (I assume that you have all these results handy since you have performed an extensive hyperparamter search). This would help to gain some better intuition for how difficult it is to tune the proposed unit. In other words, I would like to see how sharp the performance drop is if you perturb the tuning parameters slightly (i.e., plot the test accuracy as a function of \gamma and \epsilon).


Minor comments:
-------
* Given additional space, it would be nice to see an extended related work section.

* It would be nice to so results for a language modeling task.

* In Table 2, the citation for the Fast RNN is incorrect.

---

> ### Author Response · Authors · 2020-11-17
> **Reply to Reviewer 3 - Part 1**
>
> We start by thanking the reviewer for your appreciation of the merits of our paper and your welcome suggestions to improve it. We proceed to address the fair points of criticism that you raise below and thank you in advance for your patience in reading our elaborate reply.
> 1. We sincerely apologize for the unintended but natural misunderstanding caused by our original write-up about the IMEX discretization (Eqn 3). As it happens, the IMEX discretization comes in two variants that differ *only* in the discretization of the damping term $-\epsilon {\bf z}$ of (Eqn 2). As we point out in the revised version, there are two possibilities in this regard. Either we use an implicit discretization i.e. by setting $-\epsilon \bf z_n$ as the damping term or we use an explicit discretization i.e. by setting $-\epsilon \bf z_{n-1}$ as the damping term. Please note that all the other terms are still the same, including the implicit treatment of the velocity term ${\bf z}$ in the first equation of (Eqn 2). Thus, we still retain the IMEX label on the discretization as the first equation in (Eqn 2) is discretized implicitly and the second equation in (Eqn 2) is discretized explicitly. We have clearly written this in the revised version, see the sentences after (Eqn 3). We provide rigorous analysis of one of the versions in the main text i.e. the version with $-\epsilon \bf z_n$ as the damping term, and also with normalized values of the frequency and damping parameters i.e. $\gamma = \epsilon =1$. This is to simplify the proofs. However, analogous results also hold for the case where the damping term is discretized as $-\epsilon \bf z_{n-1}$ as well as for general values of $\gamma,\epsilon$. These statements and proofs are now added in **SM** section F, with a clear statement to this effect made in the line immediately after (Eqn 4) in the main text. The reviewer would appreciate that the proofs in this case are longer and more tedious (compare the proofs of proposition F.1 with that of proposition 3.1 for instance) and it is best left to put these results in **SM** while presenting the analysis for the version with $-\epsilon \bf z_{n}$ as damping term and unit values of $\gamma,\epsilon$ in the main text. On the other hand, we observed that the version with $-\epsilon \bf z_{n-1}$ provided marginally better performance on the numerical experiments and for definiteness, only provide results with it in the article. We have explicitly mentioned this in the main text in the revised version (please check the first paragraph of section 4). We hope that the reviewer is convinced by our explanation and apologize again for our unintended impression of trying to mask this issue in the original version of the manuscript. Regarding your question about implementation of both versions of the IMEX discretization, please note that the equivalent forms (Eqn 4) and (Eqn 48) of these two versions are very similar, with only differences being in scalar prefactors. Hence, it as easy to implement one as the other.
> 2. The reviewer's comments on the readability of section 3 are warranted. We have followed your very useful suggestions and added material from **SM** C in section 2 to build intuition about the underlying ODE now, including a discussion about the frequency and damping parameters. The interplay between discretization and the dynamics of the underlying ODE is also discussed in more detail. We have omitted some proofs from section 3 and moved them to the **SM**, section E.5. However, we retain a sketch of the proof for the gradient upper bound as we want to provide an argument for why the gradient norms will be close to $1$.
> 3. To address the reviewer's concern about the presented best result on sMNIST, we added to the supplementary code folder the log of the best run together with implementing the exact fixed seed and stating the pytorch version and CPU model in the readme-files in order to reproduce it. Moreover, as asked by one of the other reviewers, we have now provided the mean and standard deviation (over 10 retrainings) for all the results with coRNN, including sMNIST, in Table 5 and observe a very low standard deviation of $0.07\%$ in the sMNIST task.
> 4. The reviewer makes an excellent suggestion about an ablation study for investigating sensitivity with respect to the control parameters $\gamma,\epsilon$. We do so for the noise-padded CIFAR test case and present the results in the newly added Figure 4 (with a discussion in the main text on page 8). As observed from the figure, we see some minor variation in the test accuracy when $\epsilon,\gamma$ are perturbed from their optimal values but the maximum drop in performance is from $59\%$ to $55\%$, indicating robustness of coRNN with respect to the control parameters.
>
> Please *continue* to the second part of our reply: **Reply to Reviewer 3 - Part 2**

---

> > ### Author Response · Authors · 2020-11-17
> > **Reply to Reviewer 3 - Part 2**
> >
> > **Minor Comments.**
> > 1. We have added some further details in the extended work section and compared our approach to other ODE inspired RNNs.
> > 2. The use of coRNN on a language modeling example is indeed a very good suggestion and we intend to do it in a possible future article as the focus in this article is on learning tasks with long-term-dependencies where the mitigation of the exploding/vanishing gradient problem is of crucial importance.
> > 3. We thank the reviewer for bringing this to our attention. While we followed standard practice and cited the publication where we collected the experimental results from in table 1,2,3,4, we double-checked again that we also consistently cited the original publication of each method in the main text.

---

> > ### Comment · AnonReviewer3 · 2020-11-19
> > **All my concernes have been addressed. I will revise my rating.**
> >
> > I would like to thank the authors for the detailed response. All my concerns have been addressed and clarified and I am happy with the revised manuscript. In particular, Table 5 is convincing.  Hence, I will change my rating!
> >
> > This said, I am curious if you have some intuition for why the ablation study (Figure 4) shows that the model is largely insensitive to  $\epsilon$, but relatively sensitive to $\gamma$.

---

> > > ### Author Response · Authors · 2020-11-19
> > > **Reply to follow-up question of Reviewer 3**
> > >
> > > We would like to thank the reviewer very much for accepting our changes and increasing our score. Your question about the greater sensitivity of the results with respect to the frequency parameter $\gamma$, when compared to the damping parameter $\epsilon$ is a very interesting one. As an answer, we would like to point out that the parameters $\epsilon$ and $\gamma$ play a different role in our estimates. For instance, we see that $\epsilon$ only enters the bounds (Eqns. 50,54 and 55) indirectly through the constraint (49) on the time step. On the other hand, the hidden state and gradient bounds (Eqns. 50,54 and 55) explicitly depend on $\gamma$. This suggests the following outcome, i.e. as soon as $\epsilon$ is chosen such that the constraint on the time step is satisfied, it plays no further role on the bounds and possibly the performance of coRNN. On the other hand, $\gamma$ will play a direct role on the bounds and possibly the performance of coRNN. Thus, we might expect a much greater sensitivity of the results to $\gamma$ and not to $\epsilon$. We hope that this provides a satisfactory answer to your very valid question.

---

> ### Comment · AnonReviewer3 · 2020-11-24
> **Comparision to state-of-the-art models is missing**
>
> I just realized that the authors missed to compare their model to the Non-normal RNN [1] and Exponential RNN [2] for the add task. Both of these models are considered as state of the art for this type of problem and should be used as baseline instead of the Anitsymmetric RNN, Fast RNN and tanh RNN. In other words, the results in Figure 1 are not very compelling in their current form. This said, I am also concerned that your model will not excel on the more challenging copy task.
>
> [1] Kerg, Giancarlo, et al. "Non-normal Recurrent Neural Network (nnRNN): learning long time dependencies while improving expressivity with transient dynamics." Advances in Neural Information Processing Systems 32 (2019): 13613-13623.
>
> [2] Lezcano-Casado, Mario, and David Martínez-Rubio. "Cheap orthogonal constraints in neural networks: A simple parametrization of the orthogonal and unitary group." arXiv preprint arXiv:1901.08428 (2019).

---

> > ### Author Response · Authors · 2020-11-25
> > **Response to comment of Reviewer 3**
> >
> > We start by thanking the reviewer for your latest comment. Motivated by your question, we decided to baseline the performance of coRNN on the adding problem with yet another example of RNNs designed to learn LTDs. Given the very limited amount of time available to us for performing additional tests before the deadline of this discussion phase, we had to choose between the two architectures that you suggest, namely expRNN and non-normal RNN (nnRNN). We chose expRNN as even in the reference [1] where non-normal RNN is proposed, both the results and the discussion imply that expRNN is superior to nnRNN in learning LTDs. Thus, we tested expRNN on the adding problem (surprisingly we did not encounter any published results on either expRNN or nnRNN on the adding problem). The results are now presented in the Figure 1 of the latest version of the article. As can be clearly observed from this figure, expRNN is better performing than both FastRNN and anti-sym. RNN. However, even on the sequence length of $T=500$, coRNN readily outperforms expRNN. This difference in performance is further accentuated for a sequence length of $T=2000$, in which case the expRNN beats the baseline but does not reach a desired test MSE within training time. In comparison, coRNN converged very fast to this level of MSE. Finally on the most challenging sequence length of $T=5000$, only coRNN is able to beat the baseline whereas expRNN fails to do so. Thus, these new results clearly demonstrate the superior performance of coRNN on this particular problem when compared to state-of-the-art baselines. We emphasize that exactly the same hyperparameter selection protocol was used for all the tested architectures in order to ensure a fair comparison. We expect that nnRNN will perform similarly to expRNN for the adding problem and we would be happy to add the results in the camera-ready version of the article in case it is accepted for publication. Regarding the copying task, we had to make a choice for the experiments that we presented in the article and we felt that the adding problem was a more reasonable choice to test the performance of coRNN as it involves both memory and computation in contrast to the copying task where only the ability of an RNN to memorize is tested. We sincerely hope that we have adequately addressed the reviewer's concern.

---

### Official Review · AnonReviewer2 · 2020-10-28

**Rating:** 8
**Confidence:** 5

**Review:**

The paper introduces a novel recurrent neural network architecture which approximately preserves the norm of the gradient irrespective of the number of unroll steps. This complements a rapidly growing line of research that aims to better understand dynamical properties of RNNs and their gradients, thus potentially enabling training of models that capture long term dependencies while avoiding exploding and vanishing gradients.

The submission has the following strengths to it:

1. It offers a clear and succinct proof of the gradient stability as well as the stability of the forward dynamics.
1. It provides convincing experiments on relevant datasets, all while showing competitive results.
1. It's exceedingly well written, and readily understandable even without prior knowledge.

##### I firmly believe that based on those, it should be accepted. It has all the hallmarks of a good, paper with potentially wide application.

That being said, I do have some minor objections:

1. When presenting Eq. (1), the 'intuitive' interpretation of $\gamma, \epsilon$ should given right away, rather than deferred to later sections of the paper, especially the appendix.
1. The motivation for the use of non-linear oscillators is well-written but perhaps should be de-emphasized. I would like to put-forward the following argument for it. It appears that the choice of the dynamics only constitutes half of the 'puzzle'.
The choice of IMEX is mentioned *en passant*, but seems to be rather crucial to obtaining the theoretical guarantees viz. Proposition 3.2 and 3.3. I did not have time to check the derivation against other schemes, but I presume that the choice of IMEX was highly non-trivial in designing the new architecture. If that is indeed the case, then the role of the solver ought to be emphasized.
1. I appreciate the authors comments regarding the expressivity of the proposed architecture, as well as the demonstrations in Appendix B.
However, I would also appreciate a simple example of the kind of dynamics where the coRNN cell ought to break down -- given the corollary of Proposition 3.1, it would be interesting to show the potential break-down of the network on task that involves approximating a chaotic dynamical systems. In particular, it would be very interesting how coRNN fares against similarly sized gated RNN (LSTM or GRU).
1. For my own understanding, is the exponent $r$ in Eq. 8 there only to conveniently related $\eta$ to $\Delta t$? If not, does it admit some more intuitive interpretation?
1. Since, the proposed architecture requires a sufficiently small step-size, are the resultant equivalent (in some suitable sense be it topologically, having the same invariant set) to the continuous time dynamics?
1. Lastly, the authors report the hidden unit dimensionality, but from the main text it is entire unclear whether that's the dimensionality of $y$ or $ \left[y^\top, z^\top \right]^\top$. Having looked at the code, it appears to be the former.

#### Edit:
Out of curiosity I ran the submitted code for the permuted sequential MNIST task, and noticed that the following:
1. The numbers that the authors report in the paper seem to be result of a single network realization. While granted, this is somewhat consistent with practices common in the community, it makes one question how representative they are of different initial seeds. In the current setting it's hard to disambiguate whether the random seed was chosen coincidentally or rather specifically because of the purported state-of-the-art outcome.
1. For this reason I suggest the authors compute additional iterates of the model and report some distributional information about the best loss/accuracy for all the tasks covered in the submission.
1. Attaining "state-of-the-art" results is notable, but is by no means pre-requisite for this to be considered a good submission.

---

> ### Author Response · Authors · 2020-11-17
> **Reply to Reviewer 2 - Part 1**
>
> We start by thanking the reviewer for your appreciation of the merits of our paper and your welcome suggestions for improving it. We proceed to address each of the points that you raise below and thank you in advance for your patience in reading our elaborate reply.
> 1. We completely agree with the reviewer's point about the parameters $\gamma,\epsilon$. Based on your suggestions and that of the other reviewers, we have modified the write-up somewhat to provide an *intuitive* explanation of what our ODE system (Eqn 1) signifies. This is now added on page 2, in the first two paragraphs of the Motivation and Background section, with a clear interpretation of $\epsilon$ and $\gamma$ provided there. We have also labelled them appropriately, right after introducing them in (Eqn 1).
> 2. The reviewer makes an excellent suggestion. Indeed, your interpretation of the interplay between continuous dynamics and the choice of discretization is spot on. As we really liked this take on coRNN, we have modified our write-up to reflect it. To this end, we have added a paragraph at the beginning of section 3, where we state that the dynamics of the ODE (Eqn 2) is such that the hidden states and hidden state gradients are bounded (the precise statements and proofs are provided in the newly added **SM**, section D). Given this fact, a *structure preserving discretization* of (Eqn 2) such as (Eqn 3) would be expected to inherit these bounds and we indeed show that this is the case in section 3 and **SM** E. However, the asymptotic formula (Eqn 16) that mitigates the vanishing gradient problem is much more subtle. It seems to rely on the discretization and does not appear to have a continuous analogue. So, we have remarked this in the last paragraph of section 3 (page 5). We hope that the current write-up clearly reflects this interplay and balance between continuous dynamics and the discretization and thank you again for your suggestion.
> 3. The reviewer points to something very pertinent and correctly identifies a possible limitation of coRNN. As you state, the design of coRNN is such that chaotic behavior for the underlying ODE (Eqn 2) and the discretization is ruled out. We present the corresponding statements in **SM**, proposition D.2 and E.1. Thus, one can expect that coRNN does not show adequate performance on predicting chaotic time series. We followed your suggestion and performed an experiment with a Lorenz-96 chaotic dynamical system and present the results in **SM**, section A, see Table 6 from which we observe that coRNN is comparable in performance to LSTM when the dynamics is not yet chaotic but is clearly inferior in the chaotic regime. We have added a paragraph in the discussion (paragraph 3 on page 9), about the possible failure of coRNN in processing chaotic dynamics.
> 4. The reviewer is completely correct in asserting that the exponent $r$ serves to relate $\eta$ to $\Delta t$.
> 5. The reviewer's point about the equivalence of the long-time limit (steady state dynamics) of the underlying ODE (Eqn 2) and the discretization (Eqn 3) is correct. To see this, we formally let ${\bf y}^{\prime} = {\bf z}^{\prime}=0$ in (Eqn 2) and observe that the steady states are given as roots of the equation $\sigma({\bf W}{\bf y}+{\bf V}{\bf u} +{\bf b}) = \gamma {\bf y}$. Setting $\bf y_{n} = \bf y_{n-1}$ , $\bf z_{n} = \bf z_{n-1}$ in (Eqn 3) in order to recover the steady state dynamics, leads to exactly the same equation for steady states. However, we refrain from commenting on this aspect in the main text as we do not yet see how this equivalence has a bearing on the functioning of coRNN.
> 6. The reviewer is right about the fact that hidden unit dimensionality only refers to the dimensionality of the hidden state ${\bf y}$. We have made this explicit in the text in the revised version.
>
> Please *continue* to the second part of our reply: **Reply to Reviewer 2 - Part 2**

---

> > ### Author Response · Authors · 2020-11-17
> > **Reply to Reviewer 2 - Part 2**
> >
> > **Suggestions for Edit**.
> > * (1,2). We completely agree with the reviewer's point about the need to provide distributional information on results in order to check the robustness of the proposed network with respect to random initializations and perturbations. As the reviewer correctly points out, we have followed widely prevalent practice in the community to report *best performing* results with coRNN for each learning task. This allows us to compare with other methods as baselines, as most of the published papers only provide *best performing* results and seldom provide distributional information. Hence, we retain the *best results* with coRNN in Tables 1,2,3,4 in order to enable the reader to compare the performance of coRNN with baselines. On the other hand, we follow your suggestion and compute distributional information on coRNN for each learning task. This is now provided in Table 5, where the mean and standard deviation, over $10$ retrainings, of test accuracy with respect to all the learning tasks is reported. As observed from the table, coRNN is quite robust in its performance, with the mean being close to the best results and low standard deviations. The standard deviation is somewhat higher for psMNIST. This is not surprising as the random perturbation also varies in each of these runs.
> > * (3). The reviewer's point about "Attaining "state-of-the-art" results is notable, but is by no means pre-requisite for this to be considered a good submission" is completely justified. We agree with this assertion and prefer to tone down the phrases *comparable to* or *better than* state of the art in many places in the text, by replacing it with *competitive*. We submit that our rationale for this article is contained in the last sentence of the first paragraph of page 9 i.e. "Thus, we provide a novel and promising strategy for designing RNN
> > architectures that are motivated by the functioning of natural systems, have rigorous bounds on hidden
> > state gradients and are robust, accurate, straightforward to train and cheap to evaluate."

---

### Official Review · AnonReviewer4 · 2020-11-02
**Prevention of exploding/vanishing gradients in novel RNN architecture CorNN**

**Rating:** 7
**Confidence:** 3

**Review:**

The paper proposes a novel RNN architecture (CorNN) to tackle the infamous problem of vanishing and exploding gradients in RNNs. The novel CorNN architecture is based on time-discretized forced coupled damped nonlinear oscillators. For the gradient norm of CorNN analytical lower and upper bounds are calculated implying that CorNN avoids vanishing and exploding gradients.
This is accompanied by numerical results (including code) that demonstrate the improved trainability of CorNN in permuted sequential MNIST, and adding task and noise-padded CIFAR10 compared to some other RNN architectures (GRU, LSTM, antisymmetricRNN, IMDB sentiment analysis, and a human activity recognition task.

In summary, the paper proposes a useful and mathematically transparent way of tackling the challenge of training RNN on tasks with long-time dependencies.

Weak points of the paper:
* In the mathematical analysis the paper claims to "rigorously prove precise bounds on [...] gradients, enabling the solution of the exploding and vanishing gradient problem". If I am not misunderstanding, the mathematical bounds are actually only shown for the initial gradient norm (plus some number of steps afterward in section C4 of the Supplementary Material).

* A more in-depth comparison of numerical gradient norms and their respective upper/lower analytical bounds would be desirable, as the lower bound on the gradient in (16) is only given in as O(∆t^(3/2)) without any prefactors.

* While the numerical results demonstrate improved trainability and performance on a number of tasks, the underlying reasons remain mostly unclear. Supplementary Material B gives some heuristics on how a superposition of forced coupled damped nonlinear oscillator can generate complex output, but it doesn't explain the avoidance of exploding and vanishing gradients and the superiority to other solutions (e.g. gated units like LSTM or GRU).

* Missing: What are the limitations of CorNN, when would you expect them to fail?

* The link to biological networks is not yet very convincing. While there exist without doubt many oscillations on different scales in the brain, it is not clear how the insights gained here could be applied to the brain.

Some smaller comments:
* How could expressivity be quantified?
* figure 1: Plotting MSE with a logarithmic or semilogarithmic axis would help to distinguish small errors.
* figure 3: It would help to also plot this for other tasks.
* figure 3 lines for other tasks would be helpful

---

> ### Author Response · Authors · 2020-11-17
> **Reply to Reviewer 4 - Part 1**
>
> We start by thanking the reviewer for appreciating the merits of our paper and the welcome suggestions to improve it. We proceed to address each of the weak points that you have identified and thank you in advance for your patience in reading our elaborate reply.
> 1. Regarding the reviewer's point about the fact that "the mathematical bounds are only shown for the initial gradient norm (plus a number of steps ....)", we would like to state that both the gradient upper bound (Eqn 9) and formula for lower bound (Eqn 16) indeed hold under the assumption (Eqn 8), that constrains the time step $\Delta t$ and the weight matrices ${\bf W}, {\mathcal W}$. Clearly, one can choose a time step $\Delta t << 1$ to enforce this condition initially (before training). Does this assumption hold during and after training has concluded ? As you correctly point out, we have argued in **SM** E.4, that this assumption can hold for a number of steps during training. However, this analysis uses worst-case bounds and is far from optimal in predicting the number of learning steps for which (Eqn 8) will hold. In order to investigate this issue further, we followed your excellent suggestion and computed the terms in the inequality (Eqn 8) for all the learning tasks with long-term dependencies, considered in our article. The results, presented in the newly added Figure 3, clearly demonstrate that the verifiable assumption (Eqn 8) holds in practice for the whole duration of the training. Nevertheless, your point about the phrasing is valid and we have qualified the assertion that we rigorously mitigate the exploding and vanishing gradient problem with the phrase *under verifiable hypotheses* in different places in the text. For instance, please check the new formulation in the first line of page 2 of the revised version.
> 2. The reviewer's question about determining the exact constants (prefactors) in the gradient upper bound (Eqn 9) and lower bound (Eqn 16) encouraged us to carefully track the underlying constants. We can report that we have been able to derive a precise prefactor in the gradient upper bound, see (Eqn 9). Moreover, for the formula (Eqn 16) on the gradient lower bound, the only relevant quantity is the leading order term ${\mathcal O}(\Delta t^{\frac{3}{2}})$. Here, we had worked out at least the prefactor $\hat{c}$, defined in (Eqn 16). A careful analysis of the proof revealed the source of the constants as arising from the formula (Eqn 46). As we have remarked in some detail right below this formula, i.e. on page 22 (**SM** E.5), the source of this constant lies in the fact that the energy bound (Eqn 5) is too coarse to enable us to glean information on the size of the individual elements of the hidden state vector ${\bf y}_k$, at each time step $k$. Hence, we resort to a reasonable assumption that the total energy of hidden state vector is equidistributed (upto constants) across each of the elements of this vector. Hence, it is not possible to make the constant more explicit. On the other hand, for a concrete experiment, one can in principle evaluate formula (Eqn 46) but arguably, this becomes too cumbersome.
> 3. Regarding the reviewer's point about "but it doesn't explain the avoidance of exploding and vanishing gradients and the superiority to other solutions (e.g. gated units like LSTM or GRU)", it is a very valid point of criticism. To alleviate this, we have also followed the suggestions of yours and other reviewers to better motivate the performance of coRNN. To this end, we have added statements in the main text (and a whole new section in the **SM** i.e. section D), where we present bounds on the continuous time dynamics of the underlying ODE (Eqn 1). In particular, we show that the solutions of this ODE are bounded (Proposition D.1) and also the gradients of the solution with respect to the tunable paramters is also bounded (Proposition D.3). Thus, one can expect that a *structure preserving* discretization such as the IMEX discretization (Eqn 3) will inherit these bounds and the exploding gradient problem can be alleviated. This is in marked contrast to LSTMs/GRUs where the gradients can explode. Thus, having an underlying ODE with stable dynamics explains this point. However, the vanishing gradient problem is more subtle and seems to rely on the specifics of the discretization. We also explain a possible pathway for this in terms of the fact that formula (Eqn 14) suggests that the norms on the recurrent matrices remain close to $1$, thus one can expect that the gradient does not vanish. Please check our explanations in the first and last paragraphs of section 3 as well as the remark, immediately after formula (Eqn 14).
>
> Please *continue* to the second part of our reply: **Reply to Reviewer 4 - Part 2**

---

> > ### Author Response · Authors · 2020-11-17
> > **Reply to Reviewer 4 - Part 2**
> >
> > 4. Regarding the reviewer's question: "What are the limitations of CorNN, when would you expect them to fail?", it is indeed very interesting. As we point out in the main text in the first paragraph of page 4 (and supplement this statement with a precise results in proposition D.2 and proposition E.1 of the **SM**), the structure of our proposed RNN (and the underlying ODE) rule out chaotic behavior. Thus, a clear limitation of coRNN is in the prediction of chaotic time series. This was also identified by one of the reviewers and following their suggestion, we have expanded this point in the discussion (please check the third paragraph of page 9 in the main text) and also in section A of the **SM**, where we present results with coRNN and LSTM (with the same number of parameters) for learning the chaotic trajectories of a Lorenz system. Clearly, coRNN is inferior to LSTM in the chaotic regime.
> > 5. The reviewer's point about the connection to oscillators in neurobiology is completely valid. We wish to point out that we do not claim that coRNN will shed any light on the functioning of the brain. The link is in the other direction i.e, we were motivated by the presence of networks of oscillators in brain circuits to propose coRNNs. However, the link is more in terms of an analogy. As we state in the main text, see the last sentence of the last paragraph on Motivation and Background (page 3), we abstract the idea of oscillators in brain circuits to propose coRNN based on much simpler mechanistic systems. In general, we have toned down the neurobiological connection in the revised version but prefer to keep the sentence on motivation in page 3.
> >
> > **Smaller Comments**
> > 1. We have followed fairly common terminology in the machine learning literature about *expressivity*. Regarding the reviewer's very valid question about how to measure expressivity, we did not find a clear cut answer in the literature. The most common answer that we found was in some of the more mathematically oriented papers where the authors measure expressivity in terms of the number of tunable parameters that are necessary for a neural network to approximate/predict some underlying ground truth to a certain error tolerance. However, we would not like to assert anything definitive in this regard. We have used *expressivity* loosely as the ability of a neural network to process complex inputs/outputs and have more clearly stated this in the revised version.
> > 2. We have followed very standard practice in the literature (for instance reference Arjovsky et al) to plot figure 1 in a standard format in order to enable the reader to compare our results with other published results.
> > 3. Following the reviewer's suggestion, we have updated figure 3 to present these results for all LTD tasks that we consider. Please see answer to point 1 of the major comments on this issue. We have also added a short discussion on the results of this figure on page 8.

---

### Author Response · Authors · 2020-11-17
**Reply to all the reviewers**

At the outset, we would like to thank all four reviewers for their thorough and patient reading of our article. Their fair criticism and constructive suggestions have enabled us to improve the quality of our article. A revised version of the article is uploaded. We proceed to answer the points raised by each of the reviewers individually, below.
We would also like to point out that all the references to page numbers, sections, figures, tables, equation numbers and references, refer to those in the revised version.
Yours sincerely’
Authors of "Coupled Oscillatory Recurrent Neural Network (coRNN): An accurate and (gradient) stable architecture for learning long time dependencies"

---

### Decision · Program_Chairs · 2021-01-07
**Final Decision**

**Decision:**

Accept (Oral)

**Comment:**

A novel second order nonlinear oscillator RNN architecture is proposed, analyzed, and evaluated in this paper. The results are solid and impactful. Authors and expert reviewers showed exemplary interactions with each other, improving the manuscript in significant ways. All four reviewers overwhelmingly recommended accept. I recommend that this paper be selected as an oral presentation.